# Multiclass Loss Geometry Matters for Generalization of Gradient Descent in Separable Classification

**Matan Schliserman**
Blavatnik School of Computer Science and AI
Tel Aviv University
schliserman@mail.tau.ac.il

**Tomer Koren**
Blavatnik School of Computer Science and AI
Tel Aviv University and Google Research
tkoren@tauex.tau.ac.il

## Abstract

We study the generalization performance of unregularized gradient methods for separable linear classification. While previous work mostly deal with the binary case, we focus on the multiclass setting with $k$ classes and establish novel population risk bounds for Gradient Descent for loss functions that decay to zero. In this setting, we show risk bounds that reveal that convergence rates are crucially influenced by the geometry of the *loss template*, as formalized by Wang and Scott [24], rather than of the loss function itself. Particularly, we establish risk upper bounds that holds for any decay rate of the loss whose template is smooth with respect to the $p$-norm. In the case of exponentially decaying losses, our results indicates a contrast between the $p = \infty$ case, where the risk exhibits a logarithmic dependence on $k$, and $p = 2$ where the risk scales linearly with $k$. To establish this separation formally, we also prove a lower bound in the latter scenario, demonstrating that the polynomial dependence on $k$ is unavoidable. Central to our analysis is a novel bound on the Rademacher complexity of low-noise vector-valued linear predictors with a loss template smooth w.r.t. general $p$-norms.

## 1 Introduction

The generalization properties of gradient-based learning methods, particularly in overparameterized regimes, is a central topic of study in contemporary machine learning. A key question is how unregularized gradient methods achieve good generalization despite their potential to overfit. Early work by Soudry et al. [21] demonstrated that gradient descent (GD) applied to linearly separable data with the logistic loss asymptotically converges to the max-margin solution. This result suggests that gradient descent, when properly tuned, can avoid overfitting without explicit regularization. Extensions of this result to other optimization algorithms and loss functions have further deepened our understanding of this phenomenon in various scenarios [3, 4, 12, 13, 5].

A particularly interesting regime for these investigations is multi-class classification. In this setting, Soudry et al. [21] achieved convergence to max-margin with the cross-entropy loss, and Lyu and Li [8], Lyu et al. [9] extended the results to homogeneous models and two-layer networks. More recently, Ravi et al. [15] generalized the implicit bias analysis to a broader class of exponentially tailed loss functions using the PERM framework [24], thereby bridging the binary and multi-class settings in this context.

Beyond these asymptotic results, recent work has focused on the generalization performance of gradient-based methods in finite-time regimes. In the binary classification setting, several recent works examined gradient-based methods applied to smooth loss functions that decay to zero at infinity [20, 17, 19, 23]. These results show that strong generalization without explicit regularization, even in finite time, can be achieved by gradient methods also beyond the regime of exponentially

39th Conference on Neural Information Processing Systems (NeurIPS 2025).

tailed loss functions. In terms of bounds, their results reveal that generalization performance is fully characterized by the decay rate of the loss function.

Despite these advancements in understanding gradient methods in separable classification, finite-time generalization in the multi-class setting remains rather poorly understood—even for exponentially decaying loss functions, and particularly with regard to the dependence of risk bounds on the number of classes. In particular, several fundamental questions remains open: does unregularized GD generalize well also after finite number of iterations? Does the algorithm's generalization ability extend beyond the exponential decay setting? How do the properties of a loss function influence the achievable test loss bounds? Additionally, does the sole dependence of generalization performance on the decay rate of the loss function, as observed in the binary case, also extend to multi-class classification? In fact, the first two questions were stated as open problems by Ravi et al. [15].

In this work, we address these questions by studying the finite-time generalization properties of gradient descent when applied with a multi-class loss function $\ell : \mathbb{R}^k \times k \to \mathbb{R}$. Our findings reveal key distinctions from the binary classification case. Whereas in the binary regime risk bounds depend solely on the decay rate of the loss function, we show that in the multi-class setting risk bounds crucially depend on the geometry of the multi-class loss function, as determined by the norm with respect to which it is smooth. This differs from the results of Ravi et al. [15], that suggest that all exponentially tailed loss functions behave asymptotically similarly.

The class of functions that we consider is similar to the class considered by Wang and Scott [24], who showed that in the setting of classification with $k$ classes, losses are characterized by their template: a function $\tilde{\ell} : \mathbb{R}^{k-1} \to \mathbb{R}$ that has a simpler form than the original loss function $\ell$. For multi-class classification losses with a template that is $\beta$-smooth with respect to the $L_p$ norm and decays to zero at infinity, we establish the following upper bounds on the risk of the output of gradient descent (when the step size is tuned optimally),

$$\widetilde{O}\left(\frac{\beta k^{2/p} \rho^{-1}(\epsilon/k)^2}{\gamma^2 \min\{T, n\}}\right), \tag{1}$$

where $\rho : \mathbb{R} \to \mathbb{R}$ represents the decay rate of the loss function, $k$ denotes the number of classes, $T$ is the number of gradient steps, $n$ is the sample size, and $\gamma$ is the separation margin. These results suggest that gradient descent can generalize well for a reasonable number of classes ($k \ll T, n$). As with the bounds in the binary case established in [19], the risk bounds depend on the decay rate of the function, through the expression $\rho^{-1}(\epsilon/k)$ (though here the decay function $\rho^{-1}$ is evaluated at $\epsilon/k$, compared to $\epsilon$ in the binary case).

Next, noticing the fact that our upper bounds behave differently for templates that are smooth with respect to the $L_p$-norm for sufficiently large $p$, and the case of $p = 2$, giving better generalization bounds in the former case, we establish this separation formally, by showing tight lower bounds for any decay rate in the $L_2$ regime. We also provide examples for this separation in popular loss classes.

In terms of techniques, our analysis requires some new technical tools. First, we derive a Rademacher complexity bound for multi-class losses whose templates are smooth with respect to the $L_p$ norm ($2 \le p \le \infty$), in the low-noise regime. Next, we show that the choice of step-size of gradient descent also depends on the geometry of the loss, achieving improved optimization performance as $p$ becomes larger. Putting these technical pieces together, we obtain the aforementioned risk upper bounds. We remark that this approach applies to essentially any gradient method that produces a model with low norm and low optimization error, making it applicable beyond gradient descent.

### 1.1 Summary of Contributions

To summarize, our contributions are as follows:

- Our first main result (Theorem 1) establishes an upper bound for unregularized gradient descent in separable multiclass classification for any loss function that decays to zero. Our bound suggests that the dependence on the number of classes improves as $p$ increases.

- Our second main result (Theorem 3) shows a tight lower bound for losses with templates that are smooth with respect to $L_2$ and decays to zero. Our lower bound reveals a strict separation between templates that are smooth with respect to the $L_p$-norm for sufficiently large $p$, and the case of $p = 2$, where a polynomial dependence on the number of classes is unavoidable.

- As direct applications of our general bounds, we derive upper and lower bounds for templates with several decay rates (see Section 5). For example, in the exponential rate case, our result reveals that if the template is smooth with respect to the $L_\infty$ norm, the risk bounds align with those of the binary case and depend only logarithmically on $k$; in contrast, for $p = 2$ the rate has an unavoidable linear dependence on the number of classes.

- Finally, as an additional technical contribution that underlies our analysis, we show a new upper bound on the Rademacher complexity for multi-class classification losses in the low-noise regime, where the loss template is smooth with respect to any $L_p$ norm with $p \geq 2$, refining and extending the results in [7, 16]. In particular, our assumptions apply to the template rather than the individual loss functions, which represents a new perspective (see Section 1.2 for further discussion).

Put together, our results reveal that the geometry of the loss template plays a crucial role in the generalization behavior of gradient descent. Prior work on separable classification showed that for exponentially tailed losses, gradient descent implicitly converges toward max-margin solutions [21, 15], and that in the particular case of binary classification with more general tails, generalization depends primarily on the decay rate of the loss [17, 19]. In the more general multiclass setting, our results indicate that this behavior is strongly influenced by the smoothness properties of the loss template with respect to geometries. In particular, losses with similar decay rates can induce very different generalization bounds, depending on their underlying geometry. This can serve to explain why $\ell_\infty$-smooth losses such as the cross-entropy scale more favorably with the number of classes as compared to $\ell_2$-smooth losses.

## 1.2 Additional related work

**Convergence rates for unregularized GD in separable classification.** The risk of Gradient Descent in separable classification has been extensively studied. Firstly, the asymptotic analysis in the fundamental work of Soudry et al. [21] showed an upper bound of $1/\log(T)$ for the classification error of gradient descent. Then, using a more refined analysis Shamir [20] established tight bounds on for gradient descent applied to binary cross-entropy loss. Later, Schliserman and Koren [17], Telgarsky [23], Schliserman and Koren [19] extended this analysis. Schliserman and Koren [17] showed generalization bounds for gradient-based methods with constant step sizes in using an additional self-boundedness assumption. Telgarsky [23] established a high-probability risk bound for $T \leq n$ for batch Mirror Descent with a non-constant step size for linear models. Schliserman and Koren [19] showed tight risk bounds for the binary case were given for any smooth loss decaying to zero. While all of the aforementioned work (except Schliserman and Koren [17] that discussed the particular case of the cross entropy loss), studied binary classification, in this work we address the multi-class setting and establish risk bounds applicable to any classification loss with smooth template that decays to zero, without any additional assumptions.

**Lower bounds for unregularized GD in separable classification.** There are several lower bounds in the context of binary classification. Firstly, Ji and Telgarsky [4] presented a lower bound of $\Omega(\log(n)/\log(T))$ for the distance between the output of GD and a max margin solution with the same norm. In other work, Shamir [20] proved a lower bound of $\Omega(1/\gamma^2 T)$ for the empirical risk of GD when applied to logistic loss. More recently, Schliserman and Koren [19] showed a tight lower bounds for the risk of GD, that are valid for any decay rate of the loss function. In this work, we establish the first lower bounds for unregularized GD when applied in the multi-class setting. Our lower bound is valid for losses with a template with any decay rate that is smooth with respect to the $L_2$ norm.

**Vector-valued predictors (VVPs).** Extensive research has been dedicated to understanding the sample complexity of vector-valued predictors. For the non-smooth regime with bounded domain, Maurer [11] established upper bounds scaling as $O(k)$ for Lipschitz predictors with bounded Frobenius norm. In addition, Lei et al. [6] and Zhang and Zhang [26] derived logarithmic bounds in $k$ for $\ell_\infty$-Lipschitz VVPs with arbitrary initialization. In another work, Magen and Shamir [10] studied the role of initialization and established bounds independent of $k$ when the algorithm is initialized at the origin. However, these bounds grow exponentially with the error $\epsilon$, the Lipschitz constant $L$, and the radius of the initialization ball. For lower bounds, Magen and Shamir [10] established a generalization lower bound of $\Omega(\log k)$ for convex predictors, while Schliserman and Koren [18]

improved this to match the upper bounds of Maurer [11] under the $L_2$-Lipschitz condition for the nonsmooth case. Unlike these previous studies, our work focuses on the smooth and unregularized setting, where the effective norm of the iterates and the Lipschitz constant may be depend on $k$, optionally introducing additional multiplicative factors in the bounds.

**Fast rates for VVPs.** There is a large body of work that achieves fast rates for VVPs. For example, Reeve and Kaban [16] showed Rademacher complexity bounds that are logarithmic in $k$ for smooth losses with respect to the $L_\infty$ norm with bounded domain, while Li et al. [7] provided rates linear in $k$ for $L_2$-smooth losses. Another related work is the work of Wu et al. [25] that established fast rates generalization bounds for SGD in strongly convex settings. Importantly, in this study, we show that in multi-class classification, it suffices to assume the smoothness of the template of the loss function, rather than the actual loss function, and demonstrate that this property characterizes the generalization of gradient descent in this setting. In addition, we show Rademacher complexity bounds for the general $L_p$ norm, recovering the bounds of Li et al. [7] and Reeve and Kaban [16] as special cases.

## 2 Problem Setup

We consider the following multi-class linear classification setting. Let $\mathcal{D}$ denote a distribution over pairs $(x, y)$, where $x \in \mathbb{R}^d$ is a $d$-dimensional feature vector, and $y \in [k]$ is the class index corresponding to $x$. We assume that the data is scaled such that $\|x\|_2 \leq 1$ with probability 1 with respect to $\mathcal{D}$. Our focus is on the *separable* linear classification setting with margin. Specifically, denoting the Frobenius norm of a matrix $W \in \mathbb{R}^{k \times d}$ by $\|W\|_F$ and its $j$'th row by $W^j$, we assume the following separability assumption:

**Assumption 1** (Separability). There exists a matrix $W_* \in \mathbb{R}^{k \times d}$, with rows $W_*^1, \ldots, W_*^k$, such that $\|W_*\|_F \leq 1$ and, with probability 1 over $(x, y) \sim \mathcal{D}$,

$$\forall j \in [k] \setminus \{y\} : (W_*^y - W_*^j)^\top x \geq \gamma$$

Given a multi-class loss function $\ell : \mathbb{R}^k \times [k] \to \mathbb{R}^+$, the goal is to find a model $W \in \mathbb{R}^{k \times d}$ that minimizes the (population) risk, defined as the expected loss over the distribution $\mathcal{D}$:

$$L(W) = \mathbb{E}_{(x,y)\sim\mathcal{D}}[\ell(Wx, y)].$$

For this, we use a dataset $S = \{(x_1, y_1), \ldots, (x_n, y_n)\}$ of training examples drawn i.i.d. from $\mathcal{D}$, and optimize the empirical risk:

$$\widehat{L}(W) = \frac{1}{n} \sum_{i=1}^{n} \ell(Wx_i, y_i).$$

For convenience, we define the function $\ell_y : \mathbb{R}^k \to \mathbb{R}$ as $\ell_y = \ell(\cdot, y)$. In addition, for every vector $v \in \mathbb{R}^d$, we denote its $j$'th entry by $v[j]$.

### 2.1 Loss Functions and Templates

Here we detail the class of loss functions that we consider. First, following [24], we define the template of a multi-classification loss function.

**Definition 1** (Multi-class loss template). Given a multi-class loss function $\ell : \mathbb{R}^k \times [k] \to \mathbb{R}^+$, we say that $\tilde{\ell} : \mathbb{R}^{k-1} \to \mathbb{R}$ is a template of $\ell$, if for every class $y \in [k]$, it holds that

$$\ell(\hat{y}, y) = \tilde{\ell}(D_y \hat{y}),$$

where $D_y \in \mathbb{R}^{(k-1)\times k}$ is the negative identity matrix when the $y$th row is omitted and the $y$th column is replaced by the vector that all of its entries are 1.

Note that for every vector $v$ it holds that, $D_y v = (v[y] - v[1], v[y] - v[2], \ldots, v[y] - v[k])$, where the zero entry, $v[y] - v[y]$, is omitted.

The templates considered in this work are $\beta$-smooth with respect to $L_p$ norm for $p \geq 2$, as described in the following definition.

**Definition 2** (smoothness w.r.t. $L_p$). A differentiable function $f : \mathbb{R}^d \to \mathbb{R}$ is $\beta$-smooth function w.r.t $L_p$ norm if $\|\nabla f(v) - \nabla f(u)\|_q \leq \beta \|v - u\|_p$ for all $u, v \in \mathbb{R}^d$, where $\frac{1}{q} + \frac{1}{p} = 1$.

The primary goal of this paper is to quantify how the risk bounds depend on properties of the template $\tilde{\ell}$, especially the rate at which it decays to zero as its input approaches infinity and the particular norm $L_p$ which it is smooth with respect to it. To formalize this, we use the following definition, following [19]:

**Definition 3** (Tail Function). A function $\rho : [0, \infty) \to \mathbb{R}$ is called a *tail function* if $\rho$:

   (i) is nonnegative, 1-Lipschitz, and $\beta$-smooth convex;
   (ii) is strictly decreasing and $\lim_{u \to \infty} \rho(u) = 0$;
   (iii) satisfies $\rho(0) \geq 1$ and $|\rho'(0)| \geq \frac{1}{2}$.

In addition, we can define the following class of templates,

**Definition 4** ($\rho$-Tailed Class). For a given tail function $\rho$, the class $\tilde{C}_\rho^{\beta,p}$ is defined as of all nonnegative and convex functions $\tilde{\ell} : \mathbb{R}^{k-1} \to \mathbb{R}$ such that:

   (a) $\tilde{\ell}$ is $\beta$-smooth with respect to the $L_p$ norm.
   (b) $\lim_{t \to \infty} \tilde{\ell}(tu) = 0$ for all $u \in (\mathbb{R}^+)^{k-1}$.
   (c) $\tilde{\ell}(u) \leq \sum_{j=1}^{k-1} \rho(u[j])$ for all $u \in (\mathbb{R}^+)^{k-1}$.

Now, the actual class of functions we consider is the following class, which contains multi class classification losses.

**Definition 5** ($\rho$-Tailed MCC Class). The class $C_\rho^{\beta,p}$ is defined as all loss functions $\ell : \mathbb{R}^k \times [k] \to \mathbb{R}$ for which there exists $\tilde{\ell} \in \tilde{C}_\rho^{\beta,p}$ such that $\tilde{\ell}$ is a template of $\ell$.

The vast majority of loss functions used in multi-class classification are in $C_\rho^{\beta,p}$ for some tail function $\rho$, $p$ and $\beta$. In Section 5, we detail several applications of our bounds for popular multi-class functions.

## 2.2 Unregularized Gradient Descent

In this work, we focus on standard Gradient Descent with a fixed step size $\eta > 0$, applied to the empirical risk $\widehat{L}$. The algorithm is initialized at $W_1 = 0$ and performs updates at each step $t = 1, \ldots, T$ as follows:

$$W_{t+1} = W_t - \eta \nabla \widehat{L}(W_t).$$

The algorithm outputs the final model $W_T$.

While our primary focus is on GD, the majority of our results can also be adapted to other gradient methods.

## 3 Risk Bounds for GD on Multiclass Losses

In this section we establish our upper bound for the risk of GD, when the loss function $\ell$ is taken from the class $C_\rho^{\beta,p}$. The bound appears in the following theorem,

**Theorem 1.** *Let $\rho$ be a tail function and let $\ell$ be any loss function from the class $C_\rho^{\beta,p}$. Fix $T, n$ and $\delta > 0$. Then, with probability at least $1 - \delta$ (over the random sample $S$ of size $n$), the output of GD applied on $\widehat{L}$ with step size $\eta = 1/6k^{2/p}\beta$ initialized at $W_1 = 0$ has for any $\epsilon \leq \frac{1}{2}$ such that $\eta\gamma^2 T \leq (\rho^{-1}(\epsilon/k))^2/\epsilon$, for $p \in (2, \infty)$, it holds that*

$$L(w_T) = \tilde{O}\left(\frac{\beta k^{2/p}\rho^{-1}(\epsilon/k)^2}{\gamma^2 T} + \frac{\beta k^{2/p}\rho^{-1}(\epsilon/k)^2}{\gamma^2 n}\right).$$

*In addition, if $p = \infty$,*

$$L(w_T) = \tilde{O}\left(\frac{\beta\rho^{-1}(\epsilon/k)^2}{\gamma^2 T} + \frac{\beta\rho^{-1}(\epsilon/k)^2}{\gamma^2 n}\right).$$

In the rest of the section we detail the main techniques which we use for proving Theorem 1.

## 3.1 Bounds for the Rademacher Complexity of VVPs

Firstly, we explain our main technique, which is based on local Rademacher complexity of vector-valued function classes. We first recall the definition of the Rademacher complexity (e.g., [1]).

**Definition 6** (Rademacher complexity)**.** Let $\mathcal{Z}$ be a measurable space and $\mathcal{D}$ be a distribution over $\mathcal{Z}$. Let $\mathcal{F}$ be a class of real-valued functions mapping from $\mathcal{Z}$ to $\mathcal{F}$. Given a training set $S = \{z_1, \ldots, z_n\}$ of $n$ exmples that sampled i.i.d. from $\mathcal{Z}$. The *empirical Rademacher complexity* of $\mathcal{F}$ is defined by

$$\mathfrak{R}_S (\mathcal{F}) = \mathbb{E}_\epsilon \left[ \sup_{f \in \mathcal{F}} \frac{1}{n} \sum_{i=1}^{n} \epsilon_i f(z_i) \right],$$

where $\epsilon_1, \ldots, \epsilon_n$ are i.i.d. Rademacher random variables. In addition, the *worst-case Rademacher complexity* is defined as $\hat{\mathfrak{R}}_n (\mathcal{F}) = \sup_{S \in \mathcal{Z}^n} \mathfrak{R}_S (\mathcal{F})$.

In particular, in our work, given a loss function $\ell \in C_\rho^{\beta, p}$, we are interested in bounding the worst case Rademacher complexity of the class

$$\mathcal{L}_\ell^{B,r} = \left\{ (x, y) \mapsto \ell(Wx, y) : W \in \mathbb{B}_B^{k \times d}, \widehat{L}(W) \leq r \right\}, \tag{2}$$

where $\mathbb{B}_B^{k \times d} = \{W \in \mathbb{R}^{k \times d} \mid \|W\|_F \leq B\}$. We establish the following upper bound for the worst case Rademacher complexity of $\mathcal{L}_\ell^{B,r}$,

**Lemma 1.** *Let $\rho$ be a tail function and let $\ell \in C_\rho^{\beta, p}$. Given $B, r \geq 0$, let $\mathcal{L}_\ell^{B,r}$ be as defined above. Moreover, let $M$ be such that every $f \in \mathcal{L}_\ell^{B,r}$ is bounded by $M$. Then, it holds that,*

$$\hat{\mathfrak{R}}_n \left( \mathcal{L}_\ell^{B,r} \right) = \tilde{O} \left( \sqrt{\beta r} k^{\frac{1}{p}} \frac{B + 1}{\sqrt{n}} \right).$$

For the proof of Lemma 1, we use the approach of Lei et al. [6], Reeve and Kaban [16], that given a multi class classification training set $S = \{(x_1, y_1) \ldots (x_n, y_n)\}$, define a new training set with $nk$ examples denoted as $\tilde{S}$ follows and is defined as follows

$$\tilde{S} = \{\phi_j(x_i) \mid j \in [k], \exists y_i \text{ s.t } (x_i, y_i) \in S\},$$

where $\phi_j(x) \in \mathbb{R}^{d \times k}$ is the matrix which its $j$th column is $x$ and the rest of the columns are zero. Then, it is possible to relate the covering number of $\mathcal{L}_\ell^{B,r}$, to the covering number of the following class of linear predictors when applied on $\tilde{S}$,

$$\mathcal{H}_B = \{V \mapsto \langle W, V \rangle \mid W \in \mathbb{B}_B^{k \times d}, V \in \tilde{S}\}.$$

The full proof of Lemma 1 appears in Appendix A. Notably, in contrast to those works, which uses the properties of the loss, we show that in the multi-class classification setting, it is sufficient to use the properties of the template $\tilde{\ell}$.

The next step of the proof is to use Lemma 1, to bound the difference between the empirical risk and the population risk of a specific model in multi-class losses. Such a result appears in the following theorem,

**Theorem 2.** *Let $\rho$ be a tail function and let $\ell \in C_\rho^{\beta, p}$. Given $B, r \geq 0$, Let $\mathcal{L}_\ell^{B,r}$ be as defined above. Moreover, Let $M$ be such that every $f \in \mathcal{L}_\ell^{B,r}$ is bounded by $M$. Then, for any $\delta > 0$ we have, with probability at least $1 - \delta$ over a random sample of size $n$, for any $W \in \mathbb{B}_B^{k \times d}$,*

$$L(W) = \tilde{O} \left( \widehat{L}(W) + \frac{\beta k^{\frac{2}{p}} (B + 1)^2}{n} + \frac{M}{n} \right).$$

## 3.2 Implications of Template Geometry on Optimization

Next, we discuss how the geometry of the template influences the optimization error of GD. The key insight is that while the template $\tilde{\ell}$ is $O(1)$-smooth, this smoothness does not necessarily extend to the loss function $\ell$ with respect to the model $W$. In fact, the latter is highly dependent on the geometry of the template, as formalized in the following lemma (see proof in Appendix A):

**Lemma 2.** *Let $\|x\|_2 \leq 1$, $y \in [k]$ and $\tilde{\ell} \in \tilde{C}_\rho^{\beta,p}$ for $p \geq 2$. Let $\ell_{(x,y)} : \mathbb{R}^{k \times d} \to \mathbb{R}$ be $\ell_{(x,y)}(W) = \ell_y(Wx) = \tilde{\ell}(D_y Wx)$ Then, for every $W, W' \in \mathbb{R}^{k \times d}$,*

$$\|\nabla \ell_{(x,y)}(W) - \nabla \ell_{(x,y)}(W')\|_F \leq 3\beta k^{\frac{2}{p}} \|W - W'\|_F.$$

Since the optimal step size for GD on general $\beta$-smooth functions (with respect to $W$) is approximately $\eta \approx 1/\beta$, Lemma 2 shows that the optimal step size increases with $p$. Substituting this into the convergence bound for the optimization error of GD leads to improved convergence rates as $p$ grows, as formalized in the following lemma (see proof in Appendix A),

**Lemma 3.** *Let $\rho$ be a tail function and let $\ell \in C_\rho^{\beta,p}$. Fix any $\epsilon > 0$ and a point $W_\epsilon^* \in \mathbb{R}^{k \times d}$ such that $\widehat{L}(W_\epsilon^*) \leq \epsilon$. Then, the output of $T$-iterations GD, applied on $\widehat{L}$ with step size $\eta = 1/6k^{2/p}\beta$ initialized at $W_1 = 0$ has,*

$$\widehat{L}(W_T) \leq \frac{6k^{\frac{2}{p}}\beta \|W_\epsilon^*\|^2}{T} + 2\epsilon.$$

### 3.3 Proof of Theorem 1

We are now ready to prove Theorem 1. The proof proceeds by first showing that the iterates of GD remain within a bounded region around the origin; this is established in Lemma 14 (see Appendix A). Next, we combine the bound on the generalization gap bound from with the low-noise guarantee implied by Lemma 3 to complete the argument for Theorem 1. The full proof is detailed below.

*Proof of Theorem 1.* First, let $p \in (2, \infty)$. First, for $\epsilon$ such that $\eta\gamma^2 T \leq (\rho^{-1}(\frac{\epsilon}{k}))^2/\epsilon$, we get by Lemma 14 and Lemma 12 (see Appendix A),

$$B_\epsilon := \|W_T\| \leq 2\|W_\epsilon^*\|_F + 2\sqrt{\eta\epsilon T} \leq 2\frac{\rho^{-1}(\frac{\epsilon}{k})}{\gamma} + 2\sqrt{\eta\epsilon T} \leq \frac{4\rho^{-1}(\frac{\epsilon}{k})}{\gamma}.$$

For the same $\epsilon$, by Lemmas 3 and 12,

$$r_\epsilon := \widehat{L}(W_T) \leq \frac{\|W_\epsilon^*\|^2}{\eta T} + 2\epsilon \leq 3\frac{\rho^{-1}(\frac{\epsilon}{k})^2}{\gamma^2 \eta T}.$$

Now, we denote $\mathcal{B}_\epsilon = \{W \in \mathbb{R}^{k \times d} \| W\|_F \leq B_\epsilon\}$. Moreover, by Lemma 14 and Lemmas 12 to 14 (see Appendix A, we know that, with probability 1,

$$\begin{aligned}
M_\epsilon &= \max_{W \in \mathcal{B}_\epsilon} |\ell(Wx)| = \max_{W \in \mathcal{B}_\epsilon} \tilde{\ell}(D_y Wx) \\
&\leq 2\ell_y(W_\epsilon^* x) + \beta k^{\frac{2}{p}} \max_{W \in \mathcal{B}_\epsilon} \|W - W_\epsilon^*\|_F^2 \\
&\leq 2\ell_y(W_\epsilon^* x) + 2\beta k^{\frac{2}{p}} \max_{W \in \mathcal{B}_\epsilon} \|W\|_F^2 + 2\beta k^{\frac{2}{p}} \|W_\epsilon^*\|_F^2 \\
&\leq 2\epsilon + 8\beta k^{\frac{2}{p}} \frac{\rho^{-1}(\frac{\epsilon}{k})^2}{\gamma^2} + 2\beta k^{\frac{2}{p}} \frac{\rho^{-1}(\frac{\epsilon}{k})^2}{\gamma^2} \\
&\leq 2\epsilon + 2\frac{\rho^{-1}(\frac{\epsilon}{k})^2}{\eta\gamma^2} + \frac{\rho^{-1}(\frac{\epsilon}{k})^2}{2\eta\gamma^2} \leq 5\frac{\rho^{-1}(\frac{\epsilon}{k})^2}{\eta\gamma^2}.
\end{aligned}$$

Now, by Theorem 2, for any $\delta > 0$ we have, with probability at least $1 - \delta$ over a random sample of size $n$, for any $W \in \epsilon$, there exists a constant $C > 0$ such that $C$ depends poly-logarithmically on $k, n, M_\epsilon, \beta, \frac{1}{\delta}$ and

$$\begin{aligned}
L(W) &\leq 2\widehat{L}(W) + \tilde{C}\beta k^{\frac{2}{p}} \frac{(B_\epsilon + 1)^2}{n} + \tilde{C}\frac{M_\epsilon}{n} \\
&\leq 2\widehat{L}(W) + 4\tilde{C}\beta k^{\frac{2}{p}} \frac{B_\epsilon^2}{n} + \tilde{C}\frac{M_\epsilon}{n} \\
&\leq 2\widehat{L}(W) + \frac{64\tilde{C}\beta k^{\frac{2}{p}}\rho^{-1}(\frac{\epsilon}{k})^2}{\gamma^2 n} + \frac{5\tilde{C}\rho^{-1}(\frac{\epsilon}{k})^2}{\eta\gamma^2 n}.
\end{aligned}$$

For $W_T$ by the choice of $\eta$, we get,

$$L(W_T) \le 6\frac{\rho^{-1}(\frac{\epsilon}{k})^2}{\gamma^2\eta T} + \frac{64\tilde{C}\beta k^{\frac{2}{p}}\rho^{-1}(\frac{\epsilon}{k})^2}{\gamma^2 n} + \frac{5\tilde{C}\rho^{-1}(\frac{\epsilon}{k})^2}{\eta\gamma^2 n}$$

$$\le \frac{24\beta k^{\frac{2}{p}}\rho^{-1}(\frac{\epsilon}{k})^2}{\gamma^2 T} + \frac{84\tilde{C}\beta k^{\frac{2}{p}}\rho^{-1}(\frac{\epsilon}{k})^2}{\gamma^2 n}.$$

For $p = \infty$, since any $\beta$- smooth function w.r.t $L_\infty$ is also $\beta$ smooth with respect to the $L_k$ norm, we get that, since $x^{1/x} \le e < 3$ for any $x \in \mathbb{R}$,

$$L(W_T) \le \frac{24\beta k^{\frac{2}{k}}\rho^{-1}(\frac{\epsilon}{k})^2}{\gamma^2 T} + \frac{84\tilde{C}\beta k^{\frac{2}{k}}\rho^{-1}(\frac{\epsilon}{k})^2}{\gamma^2 n}$$

$$\le \frac{216\beta\rho^{-1}(\frac{\epsilon}{k})^2}{\gamma^2 T} + \frac{900\tilde{C}\beta\rho^{-1}(\frac{\epsilon}{k})^2}{\gamma^2 n}. \qquad \square$$

## 4 Tightness in the Euclidean case

In this section, we show that the non-trivial dependence on $k$ given in Theorem 1 for $p = 2$ is unavoidable. We prove this by establishing the following lower bound:

**Theorem 3.** *Let $p = 2$ and $\gamma \le \frac{1}{8}$. For any tail function $\rho$, sample size $n \ge 35$ and any $T$, there exist a distribution $\mathcal{D}$ and a loss function $\ell \in C_\rho^{\beta,p}$, such that for $T$-steps GD over a sample $S = \{(x_i, y_i)\}_{i=1}^n$ sampled i.i.d. from $\mathcal{D}$, initialized at $W_1 = 0$ with stepsize $\eta = 1/6\beta k$, it holds that*

$$\mathbb{E}[L(w_T)] = \Omega\left(\frac{\beta k(\rho^{-1}(\frac{256\epsilon}{k})^2}{\gamma^2 n} + \frac{\beta k(\rho^{-1}(\frac{16\epsilon}{k})^2}{\gamma^2 T}\right),$$

*for any $\epsilon < \frac{1}{256}$ such that $\eta\gamma^2 T \ge \frac{1}{\epsilon}(\rho^{-1}((\frac{\epsilon}{k})))^2$.*

For the proof of Theorem 3, we prove two lemmas. first show the following lemma, which provides a tight lower bound for the case in which $T \ge n$,

**Lemma 4.** *Let $\gamma \le \frac{1}{8}$ and $\epsilon > 0$ be such that $\frac{\rho^{-1}(\frac{\epsilon}{k})^2}{\eta\gamma^2 T} \le \epsilon \le \frac{1}{256}$. For any tail function $\rho$, sample size $n \ge 35$ and any and $T$, there exist a distribution $\mathcal{D}$ with margin $\gamma$, a loss function $\ell \in C_\rho^{\beta,p}$ for $p = 2$ such that for GD over a sample $S = \{z_i\}_{i=1}^n$ sampled i.i.d. from $\mathcal{D}$, initialized at $W_1 = 0$ with step size $\eta \le \frac{1}{6\beta k}$, it holds that*

$$\mathbb{E}[L(w_T)] = \Omega\left(\frac{\beta k\rho^{-1}(\frac{256\epsilon}{k})^2}{\gamma^2 n}\right),$$

Second, in the following lemma we give a tight lower bound for the case where $T \le n$.

**Lemma 5.** *Let $\gamma \le \frac{1}{8}$ and $\epsilon > 0$ be such that $\frac{\rho^{-1}(\frac{\epsilon}{k})^2}{\gamma^2\eta T} \le \epsilon \le \frac{1}{16}$. For any tail function $\rho$, $T$, there exist a distribution $\mathcal{D}$ with margin $\gamma$, a loss function $\ell \in C_\rho^{\beta,p}$ such that for GD over a sample $S = \{z_i\}_{i=1}^n$ sampled i.i.d. from $\mathcal{D}$, initialized at $W_1 = 0$ with step size $\eta \le \frac{1}{6\beta k}$, it holds that*

$$\mathbb{E}[L(w_T)] = \Omega\left(\frac{\rho^{-1}(\frac{16\epsilon}{k})^2}{\eta\gamma^2 T}\right),$$

Below, we provide a sketch of the proof for Lemmas 4 and 5. The full proofs and the derivation of Theorem 3 can be found in Appendix B.

To construct a hard instance for the Euclidean case and prove Lemmas 4 and 5, our main observation is that for a univariate loss function $\phi : \mathbb{R} \to \mathbb{R}$, the template $\tilde{\ell} : \mathbb{R}^{k-1} \to \mathbb{R}$, which applies $\phi$ to each entry of its input and sums the results, satisfies $\tilde{\ell} \in \tilde{C}_\rho^{\beta,p}$ for $p = 2$. This is established in the following lemma (see proof in Appendix B):

**Lemma 6.** *Let $\tilde{\ell} : \mathbb{R}^{k-1} \to \mathbb{R}$ such that there exists a function $\phi \in \mathbb{R} \to \mathbb{R}$ and $\tilde{\ell}(w) = \sum_{j=1}^{k-1} \phi(w[j])$. Then, if $\phi$ is nonnegative, convex, $\beta$-smooth and monotonically decreasing loss function such that $\phi(u) \le \rho(u)$ for all $u \ge 0$ and some function tail function $\rho$, it holds that $\tilde{\ell} \in \tilde{C}_\rho^{\beta,p}$ for $p = 2$.*

Next, to construct the hard instance, we design loss functions that represent the sum of $k$ hard binary classification instances. Combining this with a construction similar to that of Schliserman and Koren [19] for the latter case, we derive a multi-class classification lower bound for loss functions with smooth templates with respect to $L_2$.

## 5 Examples

In this section, we apply our general generalization bounds for gradient methods in the setting of multi-class classification with several popular choices of loss function, demonstrating how the geometry of the loss function affect the generalization properties of Gradient Descent.

### 5.1 Exponentially-tailed losses

First, we show a risk bound for Gradient Descent, when the decay rate of loss the loss is exponential, i.e. when $\ell \in C_\rho^{\beta,p}$ for $\rho(x) = e^{-x}$. We can apply Theorem 1 with $\epsilon = \frac{1}{T}$ and get the following,

**Corollary 4.** *Let $\ell \in C_\rho^{\beta,p}$ for $\rho(x) = e^{-x}$. Then, the output of Gradient Descent on $\widehat{L}$ with step size $\eta = \frac{1}{6k^{\frac{2}{p}}}$ and $W_1 = 0$ satisfies*

$$\mathbb{E}\left[L(W_T)\right] = \widetilde{O}\left(\frac{k^{\frac{2}{p}}}{\gamma^2 T} + \frac{k^{\frac{2}{p}}}{\gamma^2 n}\right).$$

A particular loss function in the class of losses with exponentially decaying template is the cross entropy loss, i.e., for every $y \in [k]$, $\ell_y(\hat{y}) = \log\left(1 + \sum_{j \ne y} \exp(\hat{y}[y] - \hat{y}[j])\right)$, whose template is smooth with respect to the $L_\infty$ norm (see Lemma 22 in Appendix C). Next, we can derive an upper bound for GD which is logarithmic in the number of classes. For this, we apply Theorem 1 with $\epsilon = \frac{1}{T}$ and obtain the following result,

**Corollary 5.** *If $\ell$ is the cross entropy loss function, the output of Gradient Descent on $\widehat{L}$ with step size $\eta = \frac{1}{12}$ and $W_1 = 0$ satisfies*

$$\mathbb{E}\left[L(W_T)\right] = \widetilde{O}\left(\frac{1}{\gamma^2 T} + \frac{1}{\gamma^2 n}\right).$$

This bound matches the upper bound of Schliserman and Koren [17] for Gradient Descent on the cross entropy loss, and, up to logarithmic factors matches the bounds given in Schliserman and Koren [19] for the case of setting of binary classification with smooth losses with exponential tail. In contrast, using Theorem 3 with $\epsilon = \frac{\log^2(kT)}{\eta\gamma^2 T}$ we get:

**Corollary 6.** *There exists a function $\ell \in C_\rho^{\beta,p}$ for $p = 2$ and $\rho(x) = e^{-x}$ such that the output of Gradient Descent on $\widehat{L}$ with step size $\eta = \frac{1}{6k}$ and $W_1 = 0$ holds,*

$$\mathbb{E}\left[L(W_T)\right] = \widetilde{\Omega}\left(\frac{k}{\gamma^2 T} + \frac{k}{\gamma^2 n}\right).$$

Combining Corollaries 5 and 6, we get a separation between exponentially tailed losses with templates that are smooth w.r.t the $L_\infty$-norm—such as the cross-entropy loss, where the risk matches the binary case up to logarithmic factors—and the $L_2$-norm case, the upper bounds exhibit an unavoidable linear dependence on the number of classes. This differ but not at odds with the results of [15], which suggest that exponentially tailed losses exhibit similar asymptotic behavior.

## 5.2 Polynomially-tailed losses

Now we show application of our generalization bound for Gradient Descent, when the decay rate of loss the loss is polynomial, i.e., when $\ell \in C_\rho^{\beta,p}$ for $\rho(x) = x^{-\alpha}$ for some $\alpha > 0$. For giving an upper bound for polynomially tailed losses, we can apply Theorem 1 with for this class of functions $\epsilon = \frac{k^{\frac{2}{\alpha+2}}}{(\eta\gamma^2 T)^{\frac{\alpha}{2+\alpha}}}$ and get the following upper bound,

**Corollary 7.** *Let* $\ell \in C_\rho^{\beta,p}$ *for* $\rho(x) = x^{-\alpha}$. *Then, the output of Gradient Descent on* $\widehat{L}$ *with step size* $\eta = \frac{1}{6k^{\frac{2}{p}}}$ *and* $W_1 = 0$ *holds,*

$$\mathbb{E}[L(W_T)] = \widetilde{O}\left( \frac{k^{\frac{2}{\alpha+2}\left(1+\frac{\alpha}{p}\right)}}{(\gamma^2 T)^{\frac{\alpha}{2+\alpha}}} + \frac{k^{\frac{2}{\alpha+2}\left(1+\frac{\alpha}{p}\right)} T^{\frac{2}{2+\alpha}}}{\gamma^{\frac{2\alpha}{\alpha+2}} n} \right).$$

## 6 Discussion and Limitations

In this work, we provide the first finite-time population risk bounds for gradient descent in linearly separable multiclass classification. Our results show that the geometry of the loss, captured through the $\ell_p$-smoothness of its template, plays a central role in both convergence and generalization. In contrast to prior views that emphasize the decay rate of the loss or the implicit bias of gradient methods, our analysis reveals that smoothness geometry determines how generalization of gradient descent depends on the number of classes across different multiclass regimes.

Our analysis assumes linear predictors and linearly separable data, which, while standard in theoretical studies, limits direct applicability to nonlinear or noisy settings. As a result, our results should be seen as a theoretical foundation that helps explain generalization in simpler settings, rather than a direct description of deep learning in practice. Despite these assumptions, our insights may suggest broader implications. The dependence of the bounds on $\ell_p$-smoothness offers an explanation for the empirical success of cross-entropy and other $\ell_\infty$-smooth losses in large-scale or extreme classification, where the number of classes is high.

**Future work.** Having established the first finite-time risk bounds for gradient descent in the multiclass separable setting, several open directions remain. A natural next step is to extend our analysis to nonlinear predictors and nonseparable data, and to examine empirically whether the geometric separation between smoothness norms also arises in more complex regimes. An especially relevant example is classifier-head fine-tuning in deep networks, where the data are typically nonseparable and multi-labeled, in contrast to the single-label setting considered in this work. Another promising direction is to further study the implicit bias of gradient methods for loss functions with general, potentially non-exponential tail decay rates (e.g., polynomial tails), and investigate whether it implies nontrivial multiclass risk bounds, similar to those established in this paper. This question is particularly interesting given that, in the binary case, the implicit-bias characterization of the gradient descent solutions leads to strictly suboptimal bounds as compared to the state-of-the-art [19] (see a more elaborate discussion therein).

## Acknowledgments

This project has received funding from the European Research Council (ERC) under the European Union's Horizon 2020 research and innovation program (grant agreement No. 101078075). Views and opinions expressed are however those of the author(s) only and do not necessarily reflect those of the European Union or the European Research Council. Neither the European Union nor the granting authority can be held responsible for them. This work received additional support from the Israel Science Foundation (ISF, grant numbers 2549/19 and 3174/23), a grant from the Tel Aviv University Center for AI and Data Science (TAD) and from the Len Blavatnik and the Blavatnik Family foundation.

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

# A Proofs for Section 3

We begin by the following standard lemma for smooth functions (e.g. Srebro et al. [22]).

**Lemma 7.** *Let $f : \mathbb{R}^d \to \mathbb{R}$ be a non-negative $\beta$-smooth loss function with respect to $L_p$ norm. Then, we have for every $w \in \mathbb{R}^d$,*

$$\|\nabla f(w)\|_q^2 \le 2\beta f(w, z),$$

*where $q$ is such that $\frac{1}{p} + \frac{1}{q} = 1$.*

**Lemma 8.** *Let $p \in [1, \infty]$. Let $f : \mathbb{R}^k \to \mathbb{R}$ be a non-negative $\beta$-smooth function with respect to $L_p$ norm. Then, for every $u, v \in \mathbb{R}^k$, it holds that,*

$$(f(u) - f(v))^2 \le 6\beta \max\{f(u), f(v)\}\|u - v\|_p^2.$$

*Proof.* Let $q \in [1, \infty]$ be such that $\frac{1}{p} + \frac{1}{q} = 1$. First, by the mean value theorem for any $u, v \in \mathbb{R}^k$ there exists $x$ on the line between $v$ and $u$ such that

$$0 \le f(u) - f(v) = \langle \nabla f(x), u - v \rangle$$

By smoothness, we know that

$$\|\nabla f(x) - \nabla f(v)\|_q \le \beta\|u - v\|_p.$$

As a result,

$$\|\nabla f(x)\|_q \le \|\nabla f(v)\|_q + \beta\|u - v\|_p^\alpha$$

Now, if $\|u - v\|_p \le \frac{\|\nabla f(v)\|_q}{5\beta}$ then $\|\nabla f(x)\|_q \le \frac{6}{5}\|\nabla f(v)\|_q$, by Cauchy-Schwartz inequality and Lemma 7, we get

$$
\begin{aligned}
|f(u) - f(v)|^2 &= \langle \nabla f(x), u - v \rangle^2 \\
&\le \|\nabla f(x)\|_q^2 \|u - v\|_p^2 \\
&\le \frac{36}{25}\|\nabla f(v)\|_q^2 \|u - v\|_p^2 \\
&\le 6\beta f(v)\|u - v\|_p^2 \\
&\le 6\beta \max\{f(u), f(v)\}\|u - v\|^2.
\end{aligned}
$$

Otherwise, we know that $\|\nabla f(x)\|_q \le 6\beta\|u - v\|_p$, and derive

$$
\begin{aligned}
(f(u) - f(v))^2 &= |f(u) - f(v)||\langle \nabla f(x), u - v \rangle \\
&\le |f(u) - f(v)|\|\nabla f(x)\|_q \|u - v\|_p \\
&\le 6\beta|f(u) - f(v)|\|u - v\|_p^2 \\
&\le 6\beta \max\{f(u), f(v)\}\|u - v\|_p^2.
\end{aligned}
$$

$\square$

Now, for every class $\mathcal{F}$ defined on a space $\mathcal{Z}$, $p \in [1, \infty]$, $\epsilon > 0$ and training set $S = \{z_1, \ldots, z_n\} \in \mathcal{Z}^n$, we denote by $\mathcal{N}_p(\mathcal{F}, \epsilon, n)$ the $L_p$-covering number of $\mathcal{F}$, i.e., the size of a minimal cover $C_\epsilon$ such that $\forall f \in \mathcal{F}, \exists f_\epsilon \in C_\epsilon$ s.t. $\|\tilde{f}(S) - \tilde{f}_\epsilon(S)\|_p \le \epsilon$, where for every $f \in \mathcal{F}$, $\tilde{f} : \mathcal{Z}^n \to \mathbb{R}^n$ is the function that for every set $S = \{z_1, \ldots, z_n\}$, the $i$th entry of $\tilde{f}(S)$ is $f(z_i)$.

**Lemma 9** ([22, 14, 6]). *Let $\mathcal{F}$ be a class of real-valued functions defined on a space $\tilde{\mathcal{Z}}$ and $S' := \{\tilde{z}_1, \ldots, \tilde{z}_n\} \in \tilde{\mathcal{Z}}^n$ of cardinality $n$.*

1. *If functions in $\mathcal{F}$ take values in $[-B, B]$, then for any $\epsilon > 0$ with $\text{fat}_\epsilon(\mathcal{F}) < n$ we have*

$$\log \mathcal{N}_\infty(\epsilon, \mathcal{F}, S') \le \text{fat}_\epsilon(F) \log \frac{2eBn}{\epsilon}.$$

2. *For any $\epsilon > 2\hat{\mathfrak{R}}_n(\mathcal{F})$, we have $\text{fat}_\epsilon(\mathcal{F}) < \frac{16n}{\epsilon^2}\hat{\mathfrak{R}}_n(\mathcal{F})^2$.*

3. Let $M = \sup f$. The Rademacher complexity $\Re_{S'}(\mathcal{F})$ satisfies

$$\Re_{S'}(\mathcal{F}) \leq \inf_{\xi > 0} \left( 4\xi + \frac{24}{\sqrt{n}} \int_{\xi}^{M} \sqrt{\log N_2(\epsilon, \mathcal{F}, S')} d\epsilon \right).$$

**Lemma 10.** *Let $W \in \mathbb{R}^{k \times d}$, $x \in \mathbb{R}^d$, $j \in [k]$. Then, for $\phi_j(x)$ defined in Theorem 1 it holds that,*

$$\langle W, \phi_j(x) \rangle = \langle W^j, x \rangle,$$

*where $W^j$ is the jth row of W.*

*Proof.* By the definition of $\phi_j(x)$, it holds that,

$$\langle W, \phi_j(x) \rangle = \sum_{i,j} W^j[i]\phi_j(x)^j[i]$$

$$= \sum_i W^j[i]x[i]$$

$$= \langle W^j, x \rangle$$

$\square$

**Lemma 11.** *(Proposition 7 in [6]) Let $\mathcal{H}_B$ as defined above. Then, it holds that,*

$$\hat{\Re}_{nk}(\mathcal{H}_B) \leq \frac{B}{\sqrt{nk}}$$

Now we can prove Lemma 1 and Theorem 2.

*Proof of Lemma 1.* First, notice that for every $y \in [k]$ and $v \in \mathbb{R}^k$, for every $j \neq y$, the $j$th index of $D_y v$ is $v[y] - v[j]$, we obtain that, $\|D_y v\|_\infty \leq 2\|v\|_\infty$.

Then, by Lemma 8 and the properties of $\tilde{\ell}$ we have,

$$\frac{1}{n} \sum_{i=1}^{n} (\ell_{y_i}(Wx_i) - \ell_{y_i}(W'x_i))^2 = \frac{1}{n} \sum_{i=1}^{n} (\tilde{\ell}(D_{y_i}Wx_i) - \tilde{\ell}(D_{y_i}W'x_i))^2$$

$$\leq \frac{6\beta}{n} \sum_{i=1}^{n} \left( \tilde{\ell}(D_{y_i}Wx_i) + \tilde{\ell}(D_{y_i}W'x_i) \right) \|D_{y_i}Wx_i - D_{y_i}W'x_i\|_p^2$$

$$\leq \frac{6\beta}{n} \left( \sum_{i=1}^{n} \left( \ell_{y_i}(Wx_i) + \ell_{y_i}(W'x_i) \right) \right) \left( \max_i \|D_{y_i}Wx_i - D_{y_i}W'x_i\|_p^2 \right)$$

$$\leq 12\beta r k^{\frac{2}{p}} \max_i |D_{y_i}W^j x_i - D_{y_i}W'^j x_i|_\infty^2$$

$$\leq 24\beta r k^{\frac{2}{p}} \max_i |W^j x_i - W'^j x_i|_\infty^2$$

$$\leq 24\beta r k^{\frac{2}{p}} \max_{i,j} |W^j x_i - W'^j x_i|^2$$

and get by Lemma 10

$$\sqrt{\left( \frac{1}{n} \sum_{i=1}^{n} (\ell_{y_i}(Wx_i) - \ell_{y_i}(W'x_i))^2 \right)}$$

$$\leq \sqrt{24\beta r} k^{\frac{1}{p}} \max_{i,j} |w_j x_i - w'_j x_i| = \sqrt{24\beta r} k^{\frac{1}{p}} \max_{i,j} |\langle W - W', \phi_j(x_i) \rangle|$$

We derive that

$$N_2 \left( \mathcal{L}_\ell^{B,r}, \epsilon, S \right) \leq N_\infty \left( \{ W \in \mathcal{H}_B \mid \hat{L}(W) \leq r \}, \frac{\epsilon}{\sqrt{24\beta r} k^{\frac{1}{p}}}, \tilde{S} \right) \leq N_\infty \left( \mathcal{H}_B, \frac{\epsilon}{\sqrt{24\beta r} k^{\frac{1}{p}}}, \tilde{S} \right).$$

Now by Lemma 9, for every training set $S$ it holds that

$$\Re_S \left( \mathcal{L}_\ell^{B,r} \right) \leq \inf_{\xi > 0} \left( 4\xi + \frac{24}{\sqrt{n}} \int_{\xi}^{M} \sqrt{\log N_2(\epsilon, \mathcal{L}_\ell^{B,r}, S)} d\epsilon \right)$$

$$\leq \inf_{\xi > 0} \left( 4\xi + \frac{24}{\sqrt{n}} \int_{\xi}^{M} \sqrt{\log \mathcal{N}_{\infty} \left( \frac{\epsilon}{\sqrt{24\beta r k^{\frac{1}{p}}}}, \mathcal{H}_B, \tilde{S} \right)} d\epsilon \right)$$

$$\leq \inf_{\xi \geq \sqrt{24\beta r k^{\frac{1}{p}}} \hat{\Re}_{nk}(\mathcal{H}_B)} \left( 4\xi + \frac{24}{\sqrt{n}} \int_{\xi}^{M} \sqrt{\frac{300 n k (24\beta r) k^{\frac{2}{p}}}{\epsilon^2} \hat{\Re}_{nk}(\mathcal{H}_B)^2 \log \frac{2 e M n k \sqrt{24\beta r k^{\frac{1}{p}}}}{\epsilon}} d\epsilon \right)$$

$$\leq \inf_{\xi \geq \sqrt{24\beta r k^{\frac{1}{p}}} \hat{\Re}_{nk}(\mathcal{H}_B)} \left( 4\xi + 300\sqrt{24\beta r} k^{\frac{2+p}{2p}} \hat{\Re}_{nk}(\mathcal{H}_B) \sqrt{\log \frac{2 e M n k \sqrt{24\beta r k^{\frac{1}{p}}}}{\xi}} \int_{\xi}^{M} \frac{1}{\epsilon} d\epsilon \right)$$

$$\leq \inf_{\xi \geq \sqrt{24\beta r k^{\frac{1}{p}}} \hat{\Re}_{nk}(\mathcal{H}_B)} \left( 4\xi + 300\sqrt{24\beta r} k^{\frac{2+p}{2p}} \hat{\Re}_{nk}(\mathcal{H}_B) \sqrt{\log \frac{2 e M n k \sqrt{24\beta r k^{\frac{1}{p}}}}{\xi}} \log \frac{M}{\xi} \right)$$

$$\leq 4\sqrt{24\beta r} k^{\frac{1}{p}} \frac{1}{\sqrt{n}} + 4\sqrt{24\beta r} k^{\frac{1}{p}} \hat{\Re}_{nk}(\mathcal{H}_B) +$$

$$300\sqrt{24\beta r} k^{\frac{2+p}{2p}} \hat{\Re}_{nk}(\mathcal{H}_B) \sqrt{\log \frac{\sqrt{n^3 k^2} 2 e M}{L} \hat{\Re}_{nk}(\mathcal{H}_B)} \log \frac{M \sqrt{n}}{\sqrt{24\beta r} k^{\frac{1}{p}}}$$

$$\left( \xi = \max \left\{ \sqrt{24\beta r} k^{\frac{1}{p}} \hat{\Re}_{nk}(\mathcal{H}_B), \sqrt{24\beta r} k^{\frac{1}{p}} \frac{1}{\sqrt{n}} \right\} \right)$$

$$\leq C_0 \sqrt{\beta r} k^{\frac{1}{p}} \frac{1}{\sqrt{n}} + C_1 \sqrt{\beta r} k^{\frac{2+p}{2p}} \hat{\Re}_{nk}(\mathcal{H}_B)$$

$$\leq C_0 \sqrt{\beta r} k^{\frac{1}{p}} \frac{1}{\sqrt{n}} + C_1 \sqrt{\beta r} k^{\frac{1}{p}} \frac{B}{\sqrt{n}} \qquad \text{(Lemma 11)}$$

$$\leq (C_0 + C_1) \sqrt{\beta r} k^{\frac{1}{p}} \frac{B+1}{\sqrt{n}}$$

$\square$

*Proof of Theorem 2.* By the displayed equation prior to the last one in the proof of the theorem Theorem 6.1 of [2] we have that if $\psi_n$ is any sub-root function that satisfies for all $r > 0$, $\hat{\Re}_n \left( \mathcal{L}_{\ell}^{B,r} \right) \leq \psi_n(r)$ then, for any $\delta > 0$, with probability at least $1 - \delta$, for any $W \in \mathbb{B}_B^{k \times d}$,

$$L(W) \leq \widehat{L}(W) + 45 r_n^* + \sqrt{L(W)} \left( \sqrt{8 r_n^*} + \sqrt{\frac{4M \left( \log \left( \frac{1}{\delta} \right) + 6 \log \log n \right)}{n}} \right) + \frac{20M \left( \log \left( \frac{1}{\delta} \right) + 6 \log \log n \right)}{n} \tag{3}$$

where $r_n^*$ is the largest solution to equation $\psi_n(r) = r$. Now by Lemma 1 there exists a constant $C > 0$ such that $C$ depends polylogarithmically on $k, n, M, \beta$ such that for $\psi_n(r) = C\sqrt{\beta r} k^{\frac{1}{p}} \frac{B+1}{\sqrt{n}}$, $\hat{\Re}_n \left( \mathcal{L}_{\ell}^{B,r} \right)$ satisfies the property that for all $r > 0$. Thus, for $r_n^* = C^2 \beta k^{\frac{2}{p}} \frac{(B+1)^2}{n}$ (3) holds. Now by the fact that for any non-negative $A, B, C$,

$$A \leq B + C\sqrt{A} \Rightarrow A \leq B + C^2 + \sqrt{B}C$$

we get

$$L(W) \leq \widehat{L}(W) + 106 C^2 \beta k^{\frac{2}{p}} \frac{(B+1)^2}{n} + \frac{48M}{n} \left( \log \frac{1}{\delta} + \log \log n \right) +$$

$$\sqrt{\widehat{L}(W) \left( 8 C^2 \beta k^{\frac{2}{p}} \frac{(B+1)^2}{n} + \frac{4M}{n} \left( \log \frac{1}{\delta} + \log \log n \right) \right)}$$

$$\leq \frac{3}{2} \widehat{L}(W) + 110 C^2 \beta k^{\frac{2}{p}} \frac{(B+1)^2}{n} + \frac{50M}{n} \left( \log \frac{1}{\delta} + \log \log n \right) \qquad \left( \sqrt{xy} \leq \frac{1}{2}x + \frac{1}{2}y \right)$$

$$\leq 2\widehat{L}(h) + 110 \left( \log \tfrac{1}{\delta} + \log\log\, n \right) C^2 \left( \beta k^{\frac{2}{p}} \frac{(B+1)^2}{n} + \frac{M}{n} \right). \qquad (\sqrt{xy} \leq \tfrac{1}{2}x + \tfrac{1}{2}y)$$

The theorem holds with a factor of $\tilde{C} = 110 \left( \log \tfrac{1}{\delta} + \log\log\, n \right) C^2$. $\qquad\square$

*Proof of Lemma 2.* First, similarly to Lemma 4.2 in [15], note that the expression for the gradient of $\ell_{(x,y)}$ w.r.t to $W$ is $\nabla_W \ell_{(x,y)}(W) = x \nabla \tilde{\ell}(D_y W x)^T D_y$. Let $q$ be such that $\frac{1}{p} + \frac{1}{q} = 1$. Then, it holds that

$$
\begin{aligned}
\|\nabla \ell_{(x,y)}(W) - \nabla \ell_{(x,y)}(W')\|_F^2 &= \|x(\nabla\tilde{\ell}(D_y W x) - \nabla\tilde{\ell}(D_y W' x)^T) D_y\|_F^2 \\
&= Tr\left( x(\nabla\tilde{\ell}(D_y W x) - \nabla\tilde{\ell}(D_y W' x))^T D_y D_y^T (\nabla\tilde{\ell}(D_y W x) - \nabla\tilde{\ell}(D_y W' x)) x^T \right) \\
&= Tr\left( x^T x(\nabla\tilde{\ell}(D_y W x) - \nabla\tilde{\ell}(D_y W' x))^T D_y D_y^T (\nabla\tilde{\ell}(D_y W x) - \nabla\tilde{\ell}(D_y W' x)) \right) \\
&= Tr\left( (\nabla\tilde{\ell}(D_y W x) - \nabla\tilde{\ell}(D_y W' x))^T D_y D_y^T (\nabla\tilde{\ell}(D_y W x) - \nabla\tilde{\ell}(D_y W' x)) \right) \quad (\|x\| = x^T x \leq 1) \\
&= \|D_y^T (\nabla\tilde{\ell}(D_y W x) - \nabla\tilde{\ell}(D_y W' x))\|_2^2 \\
&\leq \|D_y^T\|_{q,2}^2 \|\nabla\tilde{\ell}(D_y W x) - \nabla\tilde{\ell}(D_y W' x)\|_q^2 \\
&\leq \beta^2 \|D_y^T\|_{q,2}^2 \|D_y W x - D_y W' x\|_p^2,
\end{aligned}
$$

where for every matrix $A$, $\|A\|_{q,2}$ is $\sup_{\|v\|_q=1} \|Av\|_2$. Now, by the expression for $D_y$ it holds that, the $y$th row of $D_y^T$ is the vector with all entries as 1 and the rest of the rows with index $j$th row is a negative standard basis vector, we get that

$$
\begin{aligned}
\|D_y^T\|_{q,2}^2 &= \sup_{\|v\|_q=1} \|D_y^T v\|_2^2 \\
&\leq \sup_{\|v\|_q=1} \|v\|_1^2 + \|v_2\|_2^2 \\
&\leq 2 \left( \sup_{\|v\|_q=1} \|v\|_1 \right)^2 \\
&\leq 2k^{2(1-\frac{1}{q})} \\
&\leq 2k^{\frac{2}{p}}.
\end{aligned}
$$

Moreover, since, for every $y \in [k]$ and $v \in \mathbb{R}^k$, for every $j \neq y$, the $j$th index of $D_y v$ is $v[y] - v[j]$, we obtain that, $\|D_y v\|_\infty \leq 2\|v\|_\infty$., and First, notice that for every $y \in [k]$ and $v \in \mathbb{R}^k$, it holds that,

$$
\begin{aligned}
\|D_y v\|_p &\leq k^{\frac{1}{p}} \|D_y v\|_\infty \\
&\leq 2k^{\frac{1}{p}} \|v\|_\infty \\
&\leq 2k^{\frac{1}{p}} \|v\|_p.
\end{aligned}
$$

Then, we conclude that

$$
\begin{aligned}
\|\nabla \ell_{(x,y)}(W) - \nabla \ell_{(x,y)}(W')\|_F^2 &\leq \beta \|D_y^T\|_{q,2}^2 \|D_y W x - D_y W' x\|_p^2 \\
&\leq 8\beta^2 k^{\frac{4}{p}} \|W x - W' x\|_p^2 \\
&\leq 8\beta^2 k^{\frac{4}{p}} \|W x - W' x\|_2^2 \\
&\leq 8\beta^2 k^{\frac{4}{p}} \|W - W'\|_F^2
\end{aligned}
$$

The lemma follows by taking a square root of both sides. $\qquad\square$

**Lemma 12.** *Let $\rho$ be a tail function and let $\ell \in C_\rho^{\beta,P}$. Fix any $0 < \epsilon < \frac{1}{2}$. The, there exists a model $W_\epsilon^* \in \mathbb{R}^{k \times d}$ such that $\|W_\epsilon^*\|_F \leq \frac{\rho^{-1}(\frac{\epsilon}{k})}{\gamma}$ and $\widehat{L}(W_\epsilon^*) \leq \epsilon$.*

*Proof.* By separability, there exists a model $W^*$ such that $\|W^*\|_F \leq 1$ and such that for every $j \in [k] \setminus \{y_i\}$, it holds that $(W_*^{y_i} - W_*^j)^\top x_i \geq \gamma$ for every $(x_i, y_i)$ in the training set $S$.

Now, let $W_i^1, \ldots, W_i^{k-1} \in R^k$ be the rows of $D_{y_i} W_*$. Note that the seperability condition is equivalent to the fact that $W_i^j \cdot x_i \geq \gamma$ for any $j \in [k-1]$. Then, for $W_\epsilon^* = \frac{\rho^{-1}(\frac{\epsilon}{k})}{\gamma} W_*$ and every $(x_i, y_i) \in S$,

$$
\begin{aligned}
\ell_{y_i}(W_\epsilon^* x_i) &= \tilde{\ell}(D_{y_i} W_\epsilon^* x_i) \\
&= \tilde{\ell}\left(\frac{\rho^{-1}(\frac{\epsilon}{k})}{\gamma} D_{y_i} W_* x_i\right) \\
&\leq \sum_{j=1}^{k-1} \rho\left(\frac{\rho^{-1}(\frac{\epsilon}{k})}{\gamma} \cdot W_i^j x_i\right) \\
&\leq \sum_{j=1}^{k-1} \rho\left(\rho^{-1}(\frac{\epsilon}{k})\right) \\
&\leq \epsilon.
\end{aligned}
$$

$\square$

**Lemma 13.** *Let $y \in k$ and $\ell \in C_\rho^{\beta,p}$ for $p \geq 2$. For every $W, W' \in \mathbb{R}^{k \times d}$ such that $\|W - W'\|_F \leq R$ and $x \in \mathbb{R}^d$ with $\|x\|_2 \leq 1$, it holds that,*

$$
\tilde{\ell}(D_y W x) \leq 2\tilde{\ell}(D_y W' x) + 2\beta k^{\frac{2}{p}} R^2.
$$

*Proof of Lemma 13.* Let $W, W' \in \mathbb{R}^{k \times d}$ such that $\|W - W'\|_F \leq R$ and $x \in \mathbb{R}^d$. Moreover, Let $q$ be such that $\frac{1}{p} + \frac{1}{q} = 1$. First, notice that for every $y \in [k]$ and $v \in \mathbb{R}^k$, it holds that, $\|D_y v\|_\infty \leq 2\|v\|_\infty$. Then, by smoothness w.r.t $L_p$ and Lemma 7 it holds that

$$
\begin{aligned}
\tilde{\ell}(D_y W x) &\leq \tilde{\ell}(D_y W' x) + \nabla\tilde{\ell}(D_y W' x) \cdot (D_y W x - D_y W' x) + \frac{\beta}{2}\|D_y W x - D_y W' x\|_p^2 \\
&\leq \tilde{\ell}(D_y W' x) + \frac{1}{2\beta}\|\nabla\tilde{\ell}(D_y W' x)\|_q^2 + \frac{\beta}{2}\|W x - D_y W' x\|_p^2 + \frac{\beta}{2}\|D_y W x - D_y W' x\|_p^2 \\
&\leq 2\tilde{\ell}(D_y W' x) + \beta\|D_y W' x - D_y W x\|_p^2 \\
&\leq 2\tilde{\ell}(D_y W' x) + \beta k^{\frac{2}{p}}\|D_y W' x - D_y W x\|_\infty^2 \\
&\leq 2\tilde{\ell}(D_y W' x) + 2\beta k^{\frac{2}{p}}\|W' x - W x\|_\infty^2 \\
&\leq 2\tilde{\ell}(D_y W' x) + 2\beta k^{\frac{2}{p}} R^2,
\end{aligned}
$$

where the second inequality follows by the fact that for every $\gamma \geq 0$ and $x, y \in \mathbb{R}^k$, it holds that $xy \leq \frac{1}{2\gamma} x^2 + \frac{\gamma}{2} y^2$. $\square$

**Lemma 14.** *Fix any $\epsilon > 0$ and a point $W_\epsilon^* \in \mathbb{R}^{k \times d}$ such that $\widehat{L}(W_\epsilon^*) \leq \epsilon$. Then, the output of $T$-iterations GD, applied on $\widehat{L}$ with step size $\eta \leq 1/6k^{\frac{2}{p}}\beta$ initialized at $W_1 = 0$ has,*

$$
\|W_T - W_\epsilon^*\|_F \leq \|W_\epsilon^*\|_F + 2\sqrt{\eta\epsilon T},
$$

$$
\|W_T\|_F \leq 2\|W_\epsilon^*\|_F + 2\sqrt{\eta\epsilon T}.
$$

*Proof.* Let $\tilde{\beta} = 3k^{\frac{2}{p}}\beta$. First, by Lemma 2, $\widehat{L}$ is $\tilde{\beta}$-smooth with respect to $W$ and Lemma 7, we know that $\|\nabla\widehat{L}(W)\|^2 \leq 2\tilde{\beta}\widehat{L}(W)$ for any $W$. Therefore, by using $\eta \leq 1/\tilde{\beta}$, for every $\epsilon$,

$$
\begin{aligned}
\|W_{t+1} - W_\epsilon^*\|_F^2 &= \|W_t - \eta\nabla\widehat{L}(W_t) - W_\epsilon^*\|_F^2 \\
&= \|w_t - w_\epsilon^*\|_F^2 - 2\eta\langle W_t - W_\epsilon^*, \nabla\widehat{L}(W_t)\rangle + \eta^2\|\nabla\widehat{L}(W_t)\|_F^2 \\
&\leq \|W_t - W_\epsilon^*\|_F^2 + 2\eta\widehat{L}(W_\epsilon^*) - 2\eta\widehat{L}(W_t) + 2\tilde{\beta}\eta^2\widehat{L}(W_t) \\
&\leq \|W_t - W_\epsilon^*\|_F^2 + 2\eta\widehat{L}(W_\epsilon^*) \\
&\leq \|W_t - W_\epsilon^*\|_F^2 + 2\eta\epsilon.
\end{aligned}
$$

By summing until time $T$,

$$\|W_T - W^*_\epsilon\|^2_F \le \|W_1 - W^*_\epsilon\|^2_F + 2T\eta\epsilon = \|W^*_\epsilon\|^2_F + 2\eta\epsilon T.$$

The lemma follows by taking a square root and using triangle inequality. □

*Proof of Lemma 3.* Let $\tilde\beta = 3k^{\frac{2}{p}}\beta$. First, by Lemma 2, $\widehat{L}$ is $\tilde\beta$-smooth with respect to $W$, thus, for every $t$ and $\eta \le 1/\tilde\beta$,

$$
\begin{aligned}
\widehat{L}(W_{t+1}) &\le \widehat{L}(W_t) + \nabla\widehat{L}(W_t) \cdot (W_{t+1} - W_t) + \frac{\tilde\beta}{2}\|W_{t+1} - W_t\|^2_F \\
&= \widehat{L}(W_t) - \eta\|\nabla\widehat{L}(W_t)\|^2 + \frac{\eta^2\tilde\beta}{2}\|\nabla\widehat{L}(W_t)\|^2_F \\
&\le \widehat{L}(W_t) - \frac{\eta}{2}\|\nabla\widehat{L}(W_t)\|^2_F \\
&\le \widehat{L}(W_t).
\end{aligned}
$$

Hence,

$$\widehat{L}(W_T) \le \frac{1}{T}\sum_{t=1}^{T}\widehat{L}(W_t). \tag{4}$$

Moreover, from standard regret bounds for gradient updates, for any $W \in \mathbb{R}^{k\times d}$,

$$\frac{1}{T}\sum_{t=1}^{T}[\widehat{L}(W_t) - \widehat{L}(W)] \le \frac{\|W_1\|^2_F}{2\eta T} + \frac{\eta}{2T}\sum_{t=1}^{T}\left\|\nabla\widehat{L}(w_t)\right\|^2_F.$$

By Lemma 7,

$$\frac{1}{T}\sum_{t=1}^{T}[\widehat{L}(W_t) - \widehat{L}(W)] \le \frac{\|W\|^2_F}{2\eta T} + \frac{\eta\tilde\beta}{T}\sum_{t=1}^{T}\widehat{L}(W_t).$$

Using $\eta \le 1/2\tilde\beta$ gives

$$\frac{1}{T}\sum_{t=1}^{T}\widehat{L}(W_t) \le \frac{\|W\|^2_F}{\eta T} + 2\widehat{L}(W).$$

For $W = W^*_\epsilon$ we get by Eq. (4),

$$\widehat{L}(W_T) \le \frac{1}{T}\sum_{t=1}^{T}\widehat{L}(W_t) \le \frac{\|W^*_\epsilon\|^2_F}{\eta T} + 2\widehat{L}(W^*_\epsilon) \le \frac{\|W^*_\epsilon\|^2_F}{\eta T} + 2\epsilon.$$

When $\eta = \frac{1}{6\beta k^{\frac{2}{p}}}$, we get the lemma. □

# B   Proofs for Section 4

*Proof of Lemma 6.* The non-negativity and convexity is implied directly by the fact that $\tilde\ell$ is a sum of non-negative convex functions. Moreover, for every $u \in (\mathbb{R}^+)^k$,

$$\tilde\ell(u) = \sum_{j=1}^{k-1}\phi(u[j]) \le \sum_{j=1}^{k-1}\rho(u[j]).$$

and, since $\rho$ decays to zero at infinity

$$\lim_{t\to\infty}\tilde\ell(tu) = \lim_{t\to\infty}\sum_j\phi(tu[j]) \le \lim_{t\to\infty}\sum_j\rho(tu[j]) = \sum_j\lim_{t\to\infty}\rho(tu[j]) = 0.$$

It is left to prove the smoothness of $\tilde\ell$. For every, $u, v \in \mathbb{R}^{k-1}$, it holds that

$$\|\nabla\tilde\ell(u) - \nabla\tilde\ell(v)\|^2_2 = \sum_{i=1}^{k-1}(\phi'(u[i]) - \phi'(v[i]))^2$$

$$\leq \beta^2 \sum_{i=1}^{k-1} (u[i] - v[i])^2$$

$$= \beta^2 \|u - v\|^2.$$

$\square$

Now we turn to prove lemmas that we use in the proof of Theorem 3 We begin with probabilistic claims that similar to Schliserman and Koren [19].

**Lemma 15.** *Let $\mathcal{D}$ be the distribution defined in Eq. (5). Let $S \sim \mathcal{D}^n$ be a sample of size n, and let $(x', y') \sim \mathcal{D}$ be a validation example. Moreover, assume $n \geq 35$ and let $\delta_2$ be the fraction of $(x_2, 1)$ in S. We define the following event,*

$$A = \{(x_3, 1) \notin S\} \cap \{x' = x_3\} \cap \{\delta_2 \in [\tfrac{1}{32}, \tfrac{1}{8}]\}.$$

*Then,*

$$\Pr(A) \geq \frac{1}{120en}.$$

*Proof.* The proof follows directly from Lemma 16 and Lemma 17.

We define the following events:

$$A_1 = \{x_3 \notin S\} \cap \{x' = x_3\}, \quad A_2 = \{\delta_2 \in [\tfrac{1}{32}, \tfrac{1}{8}]\}.$$

By Lemma 16, we have

$$\Pr(A_1) \geq \frac{1}{2en}.$$

By Lemma 17, we further have

$$\Pr(A_2 \mid A_1) \geq \frac{1}{60}.$$

Combining these results, we get

$$\Pr(A) \geq \Pr(A_1) \cdot \Pr(A_2 \mid A_1) \geq \frac{1}{120en}.$$

$\square$

**Lemma 16.** *Let $\mathcal{D}$ be the distribution defined in Eq. (5). Let $S \sim \mathcal{D}^n$ be a sample of size n, and let $(x', y') \sim \mathcal{D}$ be a validation example. Let $A_1$ be the following event,*

$$A_1 = \{(x_3, 1) \notin S\} \cap \{(x', y') = (x_3, 1)\}.$$

*Then,*

$$\Pr(A_1) \geq \frac{1}{2en}.$$

*Proof.* First, observe that, since $y$ is deterministic,

$$\Pr(A_1) = \Pr(x' = x_3) \cdot \Pr((x_3, 1) \notin S \mid x' = x_3).$$

We know that

$$\Pr(x' = x_3) = \frac{1}{n}.$$

Furthermore,

$$\Pr((x_3, 1) \notin S \mid x' = x_3) = \Pr((x_3, 1) \notin S) = \left(1 - \frac{1}{n}\right)^n \geq \frac{1}{e}\left(1 - \frac{1}{n}\right) \geq \frac{1}{2e}.$$

Combining these, we obtain

$$\Pr(A_1) \geq \frac{1}{2en}.$$

$\square$

**Lemma 17.** *Let $\mathcal{D}$ be the distribution defined in Eq. (5). Assume $n \geq 35$ and let $\delta_2$ denote the fraction of $(x_2, 1)$ in S. We define the following events:*

$$A_1 = \{x_3 \notin S\} \cap \{x' = x_3\}, \quad A_2 = \{\delta_2 \in [\tfrac{1}{32}, \tfrac{1}{8}]\}.$$

*Then,*

$$\Pr(A_2 \mid A_1) = \Pr\left(\delta_2 \in [\tfrac{1}{32}, \tfrac{1}{8}] \mid A_1\right) \geq \frac{1}{60}.$$

*Proof.* For every $x_i \in S$, let $p'_i = \Pr(x_i = x_2 \mid A_1)$. Since $x_i$ and $x_j$ are independent for $i \neq j$, it follows that $p'_i = p'_j$ for all $i \neq j$. Using independence, we have:

$$p'_i = \Pr(x_i = x_2 \mid x_3 \notin S) = \Pr(x_i = x_2 \mid x_i \neq x_3).$$

This simplifies to

$$p'_i = \frac{\Pr(x_i = x_2)}{\Pr(x_i \neq x_3)} = \frac{1}{1 - \frac{1}{n}} \Pr(x_i = x_2) = \frac{5}{64}.$$

The expected value of $\delta_2$ given $A_1$ is

$$\mathbb{E}[\delta_2 \mid A_1] = \frac{1}{n} \sum_{i=1}^{n} \Pr(x_i = x_2 \mid A_1) = \frac{1}{n} \sum_{i=1}^{n} p'_i = \frac{5}{64}.$$

The variance is

$$\mathrm{Var}(\delta_2 \mid A_1) = \mathrm{Var}\left(\frac{1}{n} \sum_{i=1}^{n} \mathbb{1}_{\{x_i = x_2\}} \mid A_1\right) = \frac{1}{n^2} \sum_{i=1}^{n} \mathrm{Var}(\mathbb{1}_{\{x_i = x_2\}} \mid A_1) = \frac{5 \cdot 59}{64^2 n}.$$

Using Chebyshev's inequality, for $n \geq 35$, we have

$$\Pr(A_2 \mid A_1) = \Pr\left(\delta_2 \in [\tfrac{1}{32}, \tfrac{1}{8}] \mid A_1\right) = \Pr\left(|\delta_2 - \tfrac{5}{64}| \leq \tfrac{3}{64} \mid A_1\right).$$

Thus,

$$\Pr(A_2 \mid A_1) = 1 - \Pr\left(|\delta_2 - \tfrac{5}{64}| \geq \tfrac{3}{64} \mid A_1\right) \geq 1 - \frac{64^2}{9} \mathrm{Var}(\delta_2 \mid A_1).$$

Substituting the variance, we get

$$\Pr(A_2 \mid A_1) \geq 1 - \frac{5 \cdot 59}{9n}.$$

For $n \geq 35$, this simplifies to

$$\Pr(A_2 \mid A_1) \geq 1 - \frac{5 \cdot 59}{315} \geq \frac{1}{60}.$$

$\square$

**Lemma 18.** *Let $\rho$ be a tail function. and $\phi : \mathbb{R} \to \mathbb{R}$ be the following function*

$$\phi(x) = \begin{cases} \rho(x) & x \geq 0; \\ \rho(0) + \rho'(0)x + \frac{\beta}{2}x^2 & x < 0. \end{cases}$$

*Next, we define the following loss function for every $y$,*

$$\ell_y(\hat{y}) = \sum_{j \in [k] \setminus \{y\}} \phi(\hat{y}[y] - \hat{y}[j]).$$

*Then, $\ell \in C_\rho^{\beta, p}$.*

*Proof.* First, for $\tilde{\ell}(\hat{y}) = \sum_{j=1}^{k-1} \phi(\hat{y}_j)$, $\ell_y(\hat{y}) = \tilde{\ell}(D_y\hat{y})$. Then, it is left to prove that $\tilde{\ell} \in \tilde{C}_\rho^{\beta,p}$. By Lemma 6, it is sufficient to prove that $\phi$ is nonnegative, convex, $\beta$-smooth and monotonically decreasing loss functions such that $\phi(u) \le \rho(u)$ for all $u \ge 0$.

Second, $\phi$ is non negative: for $x \ge 0$ by the non negativity of $\rho$ and for $x < 0$ by the fact that $\rho'(0) \le 0$. Moreover, $\phi$ is convex. We need to prove that every $x < y$, $\phi'(x) \le \phi'(y)$ For $x, y < 0$, we get it by the convexity of $\rho$. For $x, y > 0$, we get it by the fact $\phi$ there is a sum of convex function and linear function. For $x < 0 < y$, by the convexity of $\rho$,

$$\phi'(x) = \rho'(0) + \beta x \le \rho'(0) \le \rho'(y).$$

In addition, $\phi$ is $\beta$-smooth. We need to prove that every $x < y$, $\phi'(y) - \phi'(x) \le \beta(y-x)$ For $x, y \ge 0$, we get it by the smoothness of $\rho$. For $x, y \le 0$, we get it by the fact that $\phi$ is a sum of $\beta$-smooth function and a linear function. For $x \le 0 \le y$, by the smoothness of $\rho$,

$$\phi'(y) - \phi'(x) = \rho'(y) - \rho'(0) - \beta x \le \beta(y-x).$$

Finally, $\phi$ is strictly monotonically decreasing. We need to prove that every $x < y$, $\phi(y) > \phi(x)$. For $x, y > 0$, we get it by the monotonicity of $\rho$. For $x < y < 0$,

$$\phi(y) = \rho(0) + \rho'(0)y + \frac{\beta}{2}y^2 \le \rho(0) + \rho'(0)x + \frac{\beta}{2}x^2 = \phi(x).$$

For $x < 0 < y$,

$$\phi(y) = \rho(y) \le \rho(0) \le \rho(0) + \rho'(0)x + \frac{\beta}{2}x^2 = \phi(x).$$

$\square$

**Lemma 19.** *Let $\phi : \mathbb{R} \to \mathbb{R}$ a univariate funcation. For every $x \in \mathbb{R}^d, y \in [k]$ and let $\ell_{x,y}$ be the following loss function*

$$\ell_{x,y}(W) = \sum_{j\in[k]\setminus\{y\}} \phi(\langle W^y - W^j, x\rangle),$$

*where for every $j$, $W^j$ is the jth row of $W$. Moreover, let $W_t$ the iterate of GD with step size $\eta > 0$, initialized on $W_1 = 0$. Then, for every $t \ge 1$, it holds that $W_t^j = W_t^2$ for any $j \ne 1$.*

*Proof.* We prove by induction on $t$. For $t = 0$, since $W_1 = 0$, the lemma trivially holds. Now, assuming $W_t^j = W_t^2$, it holds that for every $j \ne 1$, and for every possible example $x$ that the $j$th row of the gradient is $\phi'(\langle W^1 - W^j, x\rangle)x$ Then, we conclude that,

$$W_{t+1}^j = W_t^j + \eta\frac{1}{n}\sum_{i=1}^n \phi'(\langle W_t^1 - W_t^j, x_i\rangle)x_i = W_t^2 + \eta\frac{1}{n}\sum_{i=1}^n \phi'(\langle W_t^1 - W_t^2, x_i\rangle)x_i = W_{t+1}^2.$$

$\square$

*Proof of Lemma 4.* Let $\gamma \le \frac{1}{8}$. We define the following distribution $\mathcal{D}$:

$$\mathcal{D} = \begin{cases} (x_1, y_1) := ((1, 0, 0), 1) & \text{w.p. } \frac{59}{64}(1 - \frac{1}{n}); \\ (x_2, y_2) := ((-\frac{1}{2}, 3\gamma, 0), 1) & \text{w.p. } \frac{5}{64}(1 - \frac{1}{n}); \\ (x_3, y_3) := ((0, -\frac{1}{8}, 4\gamma + \frac{1}{4}), 1) & \text{w.p. } \frac{1}{n}, \end{cases} \tag{5}$$

and the following function $\phi : \mathbb{R} \to \mathbb{R}$:

$$\phi(x) = \begin{cases} \rho(x) & x \ge 0; \\ \rho(0) + \rho'(0)x + \frac{\beta}{2}x^2 & x < 0. \end{cases}$$

Then, we define the following loss function for every sample $(x, y)$,

$$\ell_y(\hat{y}) = \sum_{j\in[k]\setminus\{y\}} \phi(\hat{y}[y] - \hat{y}[j]) \tag{6}$$

First, we show that the distribution is separable. Since $y = 1$ with probability 1 for the matrix $W^*$ where its first row is $W_*^1 = (\gamma, \frac{1}{2}, \frac{1}{4})$ and for any other $j$th row $W_*^j = 0$, it holds for any $j \ne 1$ that

$(W_*^1 - W_*^j)x_i = W_*^1 x_i \geq \gamma$ for every $i \in \{1, 2, 3\}$. Moreover, Lemma 18 in Appendix B shows that indeed $\ell \in C_\rho^{\beta, p}$.

Next, let $S$ be a sample of $n$ i.i.d. examples from $\mathcal{D}$ and let $(x', y') \sim \mathcal{D}$ be a validation example independent from $S$. We denote by $\delta_2 \in [0, 1]$ the fraction of appearances of $(x_2, 1)$ in the sample $S$, and by $A_1, A_2$ the following events;

$$A_1 = \{x' = x_3 \wedge (x_3, 1) \notin S\}, \qquad A_2 = \delta_2 \in \left[\tfrac{1}{32}, \tfrac{1}{8}\right].$$

In Lemma 15 (in Appendix B), we show that

$$\Pr(A_1 \cap A_2) \geq \frac{1}{120en}. \tag{7}$$

Then by Lemma 3 and the choice of $\epsilon$,

$$\widehat{L}(W_T) \leq \frac{\rho^{-1}\left(\frac{\epsilon}{k}\right)^2}{\eta T} + 2\epsilon \leq 4\epsilon. \tag{8}$$

Now, for every $j \neq 1$, $t \in [T]$, we denote, $U_t^j = W_t^1 - W_t^j$. For the rest of the proof, we condition on the event $A_1 \cap A_2$.

First, we show that for every $j \neq 1$ it hold that $U_t^j \cdot x_2 \geq 0$. Indeed, if it were not the case, by Lemma 19, then $U_t^2 \cdot x_2 \geq 0$ and it implies that $\phi(U_t^2 \cdot x_2) > \rho(0)$; together with Eq. (8) we obtain,

$$\frac{1}{64} > 4\epsilon \geq \widehat{L}(W_T)$$

$$\geq \delta_2(k-1)\phi(U_t^2 \cdot x_2) \geq \frac{K-1}{32}\rho(0) \geq \frac{1}{32}\rho(0).$$

which is a contradiction to $\rho(0) \geq 1$. Moreover, it holds for every $j \neq 1$ that $U_T^j[1] \geq 0$. Again, we show this by contradiction for $j = 2$ and it follows for any $j \neq 1$ by Lemma 19. Conditioned on $A_2$, we have $\delta_1 > \frac{7}{8}$. Then, if $U_T^j[1] < 0$, $\phi(U_T^2 \cdot x_1) > \rho(0)$, and

$$\frac{1}{64} > 4\epsilon \geq \widehat{L}(W_T) \geq \delta_1(K-1)\phi(U_T^2 \cdot x_1) > \frac{7}{8}\rho(0).$$

which is another contradiction to the fact tat $\rho(0) \geq 1$. In addition, we notice that $x_3$ is the only possible example whose third entry is non zero. Given the event $A_1$, we know that $x_3$ is not in $S$. Equivalently, for every $(x, y) \in S$, $x[3] = 0$. As a result, since $W_1^j[3] = 0$ for every $j$, it can be proved by induction that for every $t \geq 1$, it holds for $j \neq 1$ that

$$W_{t+1}^j[3] = W_t^j + \eta \frac{1}{n}\sum_{i=1}^{n}\phi'(\langle W_t^1 - W_t^j, x_i\rangle)x_i[3] = 0.$$

For $j = 1$, it holds that,

$$W_{t+1}^j[3] = W_t^j - \frac{1}{n}\sum_{i=1}^{n}\sum_{j\neq 1}\phi'(\langle W_t^1 - W_t^j x_i\rangle)x_i[3] = 0.$$

Then, we get that for every $j \neq 1$, it holds that,

$$U_T^j \cdot x_3 = -\frac{1}{8}U_T^j(2). \tag{9}$$

Then, since we showed that $U_T^j \cdot x_2 \geq 0$ for every $j$, $\ell(W_T \cdot x_2) = \sum_{j\neq 1}\rho(U_T^j \cdot x_2)$, and conditioned on $A_2$, we have

$$32\widehat{L}(w_T)) \geq \ell(W_T \cdot x_2) = \sum_{j\neq 1}\rho(U_T^j \cdot x_2)$$

$$= (K-1)\rho(U_T^2 \cdot x_2) \geq \frac{1}{2}k\rho(U_T^2 \cdot x_2),$$

which implies for every $j \neq 1$ that,

$$U_T^j \cdot x_2 = U_T^2 \cdot x_2 \geq \rho^{-1}(\frac{64}{k}\widehat{L}(W_T)). \tag{10}$$

Therefore, by combining Eq. (10) with the fact that $U_T^j[1] \geq 0$,

$$3\gamma U_T^j[2] \geq U_T^j \cdot x_2 \geq \rho^{-1}(\frac{64}{k}\widehat{L}(W_T)).$$

This implies for every $j \neq 1$, $U_T^j[2] \geq \frac{1}{3\gamma}\rho^{-1}(\frac{64}{k}\widehat{L}(W_T))$. By Eq. (9),

$$U_T^j \cdot x_3 = -\frac{1}{8}U_T^j[2] \leq -\frac{1}{24\gamma}\rho^{-1}(\frac{64}{k}\widehat{L}(W_T)).$$

We conclude see that for every $\epsilon$ such that $\epsilon \geq \frac{(\rho^{-1}(\frac{\epsilon}{k}))^2}{\gamma^2 T \eta}$, $\widehat{L}(w_T)) \leq 4\epsilon$, and

$$\begin{aligned}
\ell(W_T x_3) = \sum_{j \neq 1} \phi(U_T^j \cdot x_3)^2 &= \sum_{j \neq 1} \rho(U_T^j \cdot x_3)^2 \\
&\geq \frac{k}{2}\frac{\beta}{2}(U_T^j \cdot x_3)^2 \geq \frac{k}{2}\frac{\beta}{2}\left(\frac{1}{24\gamma}\rho^{-1}(\frac{64}{k}\widehat{L}(w_T))\right)^2 \\
&\geq \frac{\beta k}{3000\gamma^2}\rho^{-1}(\frac{256\epsilon}{k})^2,
\end{aligned}$$

where in the final inequality we again used Eq. (8). Then the lemma follows using Eq. (7) and the law of total expectation,

$$\mathbb{E}[L(W_T)] \geq \mathbb{E}[\ell_1(w_T \cdot x_3) \mid A_1 \cap A_2]\Pr(A_1 \cap A_2). \qquad \square$$

**Lemma 20.** *Let $\rho$ be a tail function. and $\phi : \mathbb{R} \to \mathbb{R}$ be the following function*

$$\phi(x) = \begin{cases} \rho(x) & \text{if } x \geq 0; \\ \rho'(0)x + \rho(0) & \text{otherwise}. \end{cases}$$

*Next, we define the following loss function for every $y \in [k]$ ,*

$$\ell_y(\hat{y}) = \sum_{j \in [k] \setminus \{y\}} \phi(\hat{y}[y] - \hat{y}[j]).$$

*Then, $\ell \in C_\rho^{\beta,p}$.*

*Proof.* For $\tilde{\ell}(\hat{y}) = \sum_{j=1}^{k-1} \phi(\hat{y}_j)$, $\ell_y(\hat{y}) = \tilde{\ell}(D_y \hat{y})$. Then, it is left to prove that $\tilde{\ell} \in \tilde{C}_\rho^{\beta,p}$. By Lemma 6, it is sufficient to prove that $\phi$ is nonnegative, convex, $\beta$-smooth and monotonically decreasing loss functions such that $\phi(u) \leq \rho(u)$ for all $u \geq 0$.

First, $\phi$ is non negative: for $x \geq 0$ by the non negativity of $\rho$ and for $x < 0$ by the fact that $\rho'(0) \leq 0$. Moreover, $\phi$ is convex. We need to prove that every $x < y$, $\phi'(x) \leq \phi'(y)$ For $x, y < 0$, we get it by the convexity of $\rho$. For $x, y > 0$, we get it by the linearity of $\phi$. For $x < 0 < y$, by the convexity of $\rho$,

$$\phi'(x) = \rho'(0) \leq \rho'(y) = \phi'(y).$$

In addition, $\phi$ is $\beta$-smooth. We need to prove that every $x < y$, $\phi'(y) - \phi'(x) \leq \beta(y-x)$ For $x, y \geq 0$, we get it by the smoothness of $\rho$. For $x, y \leq 0$, we get it by the linearity of $\phi$. For $x \leq 0 \leq y$, by the smoothness of $\rho$,

$$\phi'(y) - \phi'(x) = \rho'(y) - \rho'(0) \leq \beta y \leq \beta(y-x).$$

Finally, $\phi$ is strictly monotonically decreasing. We need to prove that every $x < y$, $\phi(y) > \phi(x)$. For $x, y > 0$, we get it by the monotonicity of $\rho$. For $x < y < 0$,

$$\phi(y) = \rho(0) + \rho'(0)y \leq \rho(0) + \rho'(0)x = \phi(x).$$

For $x < 0 < y$,

$$\ell(y) = \rho(y) \leq \rho(0) \leq \rho(0) + \rho'(0)x = \ell(x).$$

$$\square$$

*Proof of Lemma 5.* Let $\gamma \leq \frac{1}{8}$ and $\epsilon \leq \frac{1}{16}$. We consider the following distribution;

$$\mathcal{D} = \begin{cases} (x_1, y_1) := ((1, 0), 1) & \text{with prob. } 1 - p; \\ (x_2, y_2) := ((-\frac{1}{2}, 3\gamma), 1) & \text{with prob. } p, \end{cases}$$

where $p = \frac{\rho^{-1}(\frac{16\epsilon}{k})}{72\gamma^2 Tk\eta}$. Note that by the condition of the theorem, $p \leq \epsilon \leq \frac{1}{16}$. Since $y = 1$ with probability 1 for the matrix $W_*$ where its first row is $W_*^1 = (\gamma, \frac{1}{2}, \frac{1}{4})$ and for any other $j$th row $W_*^j = 0$, it holds for any $j \neq 1$ that $\langle W_*^1 - W_*^j, x_i \rangle = \langle W_*1, x_i \rangle \geq \gamma$ for every $i \in \{1, 2\}$. In addition, we consider the following univariate function,

$$\phi(x) = \begin{cases} \rho(x) & \text{if } x \geq 0; \\ \rho'(0)x + \rho(0) & \text{otherwise.} \end{cases}$$

and the loss function such that for every $y \in [k]$,

$$\ell_y(\hat{y}) = \sum_{j \in [k]\setminus\{y\}} \phi(\hat{y}[y] - \hat{y}[j]).$$

First, by Lemma 20 we get that $\ell \in C_\rho^{\beta, p}$. Next, let $S$ be a sample of $n$ i.i.d. examples from $\mathcal{D}$. We denote by $\delta_2 \in [0, 1]$ the fraction of appearances of $(x_2, 1)$ in the sample $S$, and by $A_1$ the event that $\delta_2 \leq 2p$. By Markov's inequality, we know that $\Pr(A_1) \geq \frac{1}{2}$. Moreover, by Lemma 3 and the choice of $\epsilon$,

$$\widehat{L}(W_T) \leq 2\epsilon + \frac{2\rho^{-1}(\epsilon)^2}{\gamma^2 \eta T} \leq 4\epsilon. \tag{11}$$

By Lemma 19 we notice that all of the rows of $W_T$ except the first row are equal. Then, defining $U_T^j = W_T^1 - W_T^J$, we get that for every $j \neq 1$ it holds that $U_T^j = U_T^2$ Now, we turn to assume that $A_1$ holds. We know that

$$\delta_2 \leq 2p \leq \frac{\rho^{-1}(8\epsilon)}{36\gamma^2 T\eta} \leq \epsilon < \frac{1}{8}, \tag{12}$$

thus, conditioned on $A_1$ and by Eq. (11),

$$4\epsilon \geq \widehat{L}(W_T) > (1 - \delta_2)\ell(W_T x_1) = (1 - \delta_2) \sum_{j \neq 1} \phi(U_T^j x_1) \geq \frac{1}{2}(k - 1)\phi(U_T^2[1]). \tag{13}$$

Then, if $U_T^2[1] < 0$, we get that

$$4\epsilon > \frac{k - 1}{2}\phi(U_T^2[1]) > \frac{1}{2}\phi(0) = \frac{1}{2}\rho(0) \geq \frac{1}{2}$$

which is a contradiction to our assumption that $\epsilon \leq \frac{1}{16}$. Then $U_T^2(1) \geq 0$ and by Eq. (13), we get that $\frac{16\epsilon}{k} \geq \phi(U_T^2[1]) = \rho(U_T^2[1])$. This implies that

$$\phi(U_T^2[1]) \geq \rho^{-1}(\frac{16\epsilon}{k}). \tag{14}$$

Now, by the fact that $\rho'(0) \leq 1$ and $\rho$ is 1-Lipschitz, it follows that $\phi$ is 1-Lipschitz. Thus, by the GD update rule, it holds for every $j \neq 1$, that,

$$W_{t+1}^j[2] = W_t^j[2] + 3\eta \cdot \gamma\delta_2\phi'(\langle W_t^1 - W_t^j, x_2\rangle),$$

and for $j = 1$

$$W_{t+1}^1[2] = W_t^1[2] - 3\eta \cdot \gamma\delta_2 \sum_{j \neq 1} \phi'(\langle W_t^1 - W_t^j, x_2\rangle).$$

We get that for any $j \neq 1$,

$$U_T^j[2] \leq 3k\gamma\delta_2\eta T. \tag{15}$$

As a result, by Eqs. (12), (14) and (15) we now obtain that

$$U_T^j \cdot x_2 \leq 9k\gamma^2 \delta_2 T\eta - \frac{1}{2}\rho^{-1}(\frac{16\epsilon}{k})$$

$$\leq 9k\gamma^2 T\eta \frac{\rho^{-1}(\frac{16\epsilon}{k})}{36\gamma^2 T\eta k} - \frac{1}{2}\rho^{-1}(\frac{16\epsilon}{k})$$

$$= -\frac{1}{4}\rho^{-1}(\frac{16\epsilon}{k}).$$

By the fact that $\forall x < 0 : \phi(x) \geq -x$, this implies that in the event $A_1$ it holds that:

$$\phi(U_T^j \cdot x_2) \geq -U_T^j \cdot x_2 \geq \frac{1}{4}\rho^{-1}(\frac{16\epsilon}{k}), \tag{16}$$

and,

$$\ell_1(W_T x_2) = \sum_{j \neq 1} \phi(U_T^j \cdot x_2) \geq \frac{k}{8}\rho^{-1}(\frac{16\epsilon}{k})$$

Finally, for a new validation example $(x', y') \sim \mathcal{D}$ (independent from the sample S), $y' = 1$, and

$$\Pr(\{x' = x_2\} \cap A_1) = \Pr(x' = x_2 \mid A_1)\Pr(A_1) \geq \frac{1}{2}P(x' = x_2) = \frac{1}{2}p \geq \frac{\rho^{-1}(\frac{16\epsilon}{k})}{144\gamma^2 T\eta k}, \tag{17}$$

To conclude, from Eqs. (16) and (17) we have

$$\mathbb{E}L(W_T) \geq \mathbb{E}[\ell_1(W_T x_2) \mid \{x' = x_2\} \cap A_1]\Pr(\{x' = x_2\} \cap A_1)$$

$$\geq \frac{\rho^{-1}(\frac{16\epsilon}{k})}{144\gamma^2 T\eta k} \cdot \frac{k}{8}\rho^{-1}\left(\frac{16\epsilon}{k}\right)$$

$$\geq \frac{\rho^{-1}(\frac{16\epsilon}{k})^2}{5000\gamma^2 T\eta}.$$

$\square$

*Proof of Theorem 3.* By Lemma 4, there exists a constant $C_1$ such that $\mathbb{E}L(W_T) \geq C_1 \frac{\beta k \rho^{-1}(\frac{256\epsilon}{k})^2}{\gamma^2 n}$. By Lemma 5, there exists a constant $C_2$ such that $\mathbb{E}L(W_T) \geq C_2 \frac{\rho^{-1}(\frac{16\epsilon}{k})^2}{\eta\gamma^2 T}$. If $\frac{(\rho^{-1}(\frac{16\epsilon}{k})^2}{\gamma^2 T\eta} \geq \frac{\beta k(\rho^{-1}\frac{256\epsilon}{k})^2}{\gamma^2 n}$, the theorem follows from Lemma 5 with $\eta = \frac{1}{6\beta k}$; otherwise, it follows from Lemma 4. $\square$

## C   Proofs for Section 5

**Lemma 21.** *Let* $\alpha > 0$. *If for every* $y$, $\ell_y(\hat{y}) = \frac{1}{\alpha}\log\left(1 + \sum_{j \neq y}\exp(\alpha(\hat{y}_y - \hat{y}_j))\right)$. *Then,* $\ell \in C_\rho^{\beta,p}$ *for* $\rho(x) = \frac{1}{\alpha}e^{-\alpha x}$, $\beta = \alpha^2$ *and* $p = \infty$.

*Proof of Lemma 21.* Here we notate the $j$th entry of every vector $w$ by $w_j$.

First, for $\tilde{\ell}(\hat{y}) = \frac{1}{\alpha}\log\left(1 + \sum_{j=1}^{k-1}\exp(\alpha\hat{y}_j)\right)$, $\ell_y(\hat{y}) = \tilde{\ell}(D_y\hat{y})$. Now, $x \geq \log(1 + x) \geq 0$ for every $x$, it follows $\tilde{\ell}$ non-negative and,

$$\tilde{\ell}(\hat{y}) = \frac{1}{\alpha}\log\left(1 + \sum_{j=1}^{k-1}\exp(\alpha\hat{y}_j)\right) \leq \sum_{i=1}^{k-1}\frac{1}{\alpha}\exp(\alpha\hat{y}_j).$$

$$0 \leq \lim_{t\to\infty}\tilde{\ell}(tu) \leq lim_{t\to\infty}\sum_{i=1}^{k}\frac{1}{\alpha}\exp(\alpha t\hat{y}_j) = 0$$

For the convexity of $\tilde{\ell}$, let $u, v \in \mathbb{R}^{k-1}$ and $\lambda \in (0, 1)$. If $\tilde{u}, \tilde{v}$ are the vectors on $\mathbb{R}^k$ whose the $k - 1$ first entries are $u, v$, respectively and last entry is 0. It holds that,

$$\tilde{\ell}(\lambda u + (1 - \lambda)v) = \frac{1}{\alpha}\log\left(1 + \sum_{j=1}^{k-1}e^{\lambda\alpha u_j + (1-\lambda)\alpha v_j}\right)$$

$$= \frac{1}{\alpha} \log \left( \sum_{j=1}^{k} e^{\lambda \alpha \tilde{u}_j} e^{(1-\lambda)\alpha \tilde{v}_j} \right)$$

$$\leq \frac{1}{\alpha} \log \left( \left( \sum_{j=1}^{k} e^{\alpha \tilde{u}_j} \right)^{\lambda} \cdot \left( \sum_{j=1}^{k} e^{\alpha \tilde{v}_j} \right)^{1-\lambda} \right) \quad \text{(holder inequality for } p = \frac{1}{\lambda}, q = \frac{1}{1-\lambda})$$

$$= \frac{1}{\alpha} \lambda \log \left( \sum_{j=1}^{k} e^{\alpha \tilde{u}_j} \right) + \frac{1}{\alpha} (1-\lambda) \left( \sum_{j=1}^{k} e^{\alpha \tilde{v}_j} \right)$$

$$= \frac{1}{\alpha} \lambda \log \left( 1 + \sum_{j=1}^{k-1} e^{\alpha u_j} \right) + \frac{1}{\alpha} (1-\lambda) \log \left( 1 + \sum_{j=1}^{k-1} e^{\alpha v_j} \right)$$

$$= \lambda(\tilde{\ell}(u) + (1-\lambda)\tilde{\ell}(v),$$

as required. For the smoothness, for every $u \in \mathbb{R}^{k-1}$ the partial derivatives of $\tilde{\ell}$ are

$$\frac{\partial \tilde{\ell}}{\partial u_j}(u) = \frac{1}{\alpha} \frac{\alpha e^{\alpha u_j}}{1 + \sum_{j=1}^{k-1} e^{\alpha u_j}}$$

$$= \frac{e^{\alpha u_j}}{1 + \sum_{j=1}^{k-1} e^{\alpha u_j}}$$

$$\frac{\partial \tilde{\ell}}{\partial u_j \partial u_r}(u) = \begin{cases} \frac{-\alpha e^{\alpha u_j} e^{\alpha u_r}}{\left(1+\sum_{j=1}^{k-1} e^{\alpha u_j}\right)^2} & j \neq r \\ \frac{-\alpha e^{\alpha u_j} e^{\alpha u_j}}{\left(1+\sum_{j=1}^{k-1} e^{\alpha u_j}\right)^2} + \frac{\alpha e^{\alpha u_j}}{1+\sum_{j=1}^{k-1} e^{\alpha u_j}} & j = r \end{cases}$$

Then, if we denote by $w$ the vector that its $j$th entry is $w_j = \frac{\alpha e^{\alpha u_j}}{1+\sum_{j=1}^{k-1} e^{\alpha u_j}}$, it holds that $\nabla^2 \tilde{\ell}(w) = diag(w) - ww^T$. Now, let $v \in \mathbb{R}^{k-1}$. For $L_\infty$ smoothness it is sufficient to prove that $v^T \nabla^2 \tilde{\ell}(u)v \leq \alpha^2 \|v\|_\infty^2$.

$$v^T \nabla^2 \tilde{\ell}(u)v = v^T (diag(w) - ww^T)v$$

$$= v^T diag(w)v - (w^T v)^2$$

$$\leq v^T diag(w)v$$

$$\leq \sum_{j=1}^{k-1} w_j v_j^2$$

$$\leq \|v\|_\infty^2 \alpha \sum_{i=1}^{k-1} w_i$$

$$\leq \alpha^2 \|v\|_\infty^2.$$

$\square$

**Lemma 22.** *If $\ell$ is the cross entropy loss function, $\ell \in C_\rho^{\beta,p}$ for $\rho(x) = e^{-x}$, $\beta = 1$ and $p = \infty$.*

*Proof of Lemma 22.* The proof is implied directly from Lemma 21 with $\alpha = 1$. $\square$

