# OpenReview forum: "Multiclass Loss Geometry Matters for Generalization of Gradient Descent in Separable Classification"
_NeurIPS.cc/2025/Conference — NeurIPS 2025 poster_

### Official Review · Reviewer_m1mM · 2025-06-30

**Clarity:** 3
**Significance:** 2
**Originality:** 2
**Rating:** 4
**Confidence:** 4

**Summary:**

This paper investigates the generalization performance of (early-stopped) gradient descent for linear multiclass classification under the assumption of separability. The authors analyze a family of loss functions characterized by a tail-decay condition and a smoothness condition. A key aspect of their analysis is the careful tracking of the dependency on the number of classes, k, which has often been a subtle but important detail in reducing multiclass problems to binary ones.

The central contribution is an excess risk upper bound that explicitly details the influence of k. The authors reveal an interesting insight: the risk's dependence on k is tied to the geometry of the loss function, specifically the norm with respect to which it is smooth, in the worst case. For exponentially-tailed losses that are also smooth with respect to the L-infinity norm, the risk exhibits a mild logarithmic dependence on k. In contrast, for exponentially-tailed losses that are only smooth with respect to the L2-norm, the risk can degrade to a polynomial dependence on k in the worst case.

The theoretical analysis builds upon established techniques, including Rademacher complexity, optimistic rates, and variants of GD analysis for binary logistic regression. The primary technical novelty is a careful calculation of the Rademacher complexity in the multiclass setting. The lower bound, which is specific to the p=2 case, is derived by extending prior results for binary classification by Schliserman and Koren.

**Questions:**

Please see above.

**Ethical Concerns:**

["NO or VERY MINOR ethics concerns only"]

**Final Justification:**

I maintain my initial rating. I'll support the paper as a poster presentation. I think the results aren’t quite novel enough to warrant a spotlight or oral presentation given the prior results in the binary case.

**Limitations:**

Please see above.

**Paper Formatting Concerns:**

I don't have concerns regarding the formatting.

**Quality:**

2

**Strengths And Weaknesses:**

Overall, I find this submission in the boardline case. I am leaning towards accepting it as there is some interesting new knowledge, but the contribution is not novel enough for a clear acceptance. I wouldn’t be too sad if the paper is not accepted.

### Questions and Suggestions for the Authors

On the Optimality of the Stepsize: The discussion regarding the stepsize in Line 233, which claims that \eta = 1/\beta is "optimal," appears to overlook recent findings. For instance, in the binary separable case, the work by [Wu, Bartlett, Telgarsky, Yu, 2024, Large Stepsize Gradient Descent for Logistic Loss: Non-Monotonicity of the Loss Improves Optimization Efficiency] has demonstrated that the stepsize can be significantly larger than 1/\beta, which improves optimization and sometimes doesn’t hurt generalization. Given that this work also operates in a separable regime, it is likely that a similar phenomenon occurs here. I would encourage the authors to revisit this claim and either provide justification for why 1/\beta is optimal in the multiclass setting or revise the discussion to reflect a more nuanced understanding of the stepsize's role.

Illustrative Example for Corollary 6: Corollaries 5 and 6 effectively highlight the contrasting behaviors of different loss functions. To make the implications of Corollary 6 more concrete, it would be highly beneficial to provide an example of a specific loss function that satisfies the conditions for the worse polynomial dependence on k. This would offer readers a clearer intuition for what constitutes a "bad" loss function in this context and strengthen the practical relevance of the theoretical finding.

### Minor Point

Lemma 1: The constant M is introduced in the statement of Lemma 1 but does not appear in its conclusion. It could be removed for clarity.

---

> ### Author Rebuttal · Authors · 2025-07-28
>
> Thank you for the feedback! Below we address all of your comments; if anything remains unclear, please let us know and we will happily elaborate during the discussion period.
>
>
> >”The discussion regarding the stepsize in Line 233, which claims that $\eta = 1/\beta$ is "optimal," appears to overlook recent findings.”
>
> This is a very good point. Our understanding is that the work [Wu, Bartlett, Telgarsky, Yu, 2024, Large Stepsize Gradient Descent for Logistic Loss: Non-Monotonicity of the Loss Improves Optimization Efficiency] discusses the specific case of the logistic loss, while our work give upper and lower bounds for general functions with $\beta$-smooth template.
> As a result, the word “optimal” in line 233 referred to the optimal choice for general smooth functions, which is $\Theta(1/\beta)$. For the latter, it is a folklore fact that even for $\eta=2/\beta$, GD might not converge to a local minimum. We will clarify this point in the final version of the paper.
>
>
> >”it would be highly beneficial to provide an example of a specific loss function that satisfies the conditions for the worse polynomial dependence on k”
>
> Corollary 6 is a direct application of Lemmas 4 and 5. As a result, as detailed in the proof of Theorem 3  the function that achieves the lower bound is $\ell(Wx)=\frac12 \ell_4(Wx)+\frac12 \ell_5(Wx)$, where $\ell_4$ and $\ell_5$ are the functions from Lemmas 4 and 5 with $\rho(x)=e^{-x}$. We will add a proof for Corollary 6 in the next revision of the paper.

---

> > ### Comment · Reviewer_m1mM · 2025-08-04
> > **Comment**
> >
> > Thanks for the clarification. I’m good with the question as it stands. I'll support the paper as a poster presentation. I think the results aren’t quite novel enough to warrant a spotlight or oral presentation given the prior results in the binary case.

---

### Official Review · Reviewer_WLTb · 2025-07-02

**Clarity:** 4
**Significance:** 3
**Originality:** 3
**Rating:** 5
**Confidence:** 3

**Summary:**

This work explores the generalization of unregularized gradient methods for separable linear classification. Notably, this work examines the multiclass setting, which is often overlooked in the literature. In the finite-time setting, this work identifies unique behaviors in the multi-class setting compared to the binary setting.

**Questions:**

1. Could you comment on these insights generalizing beyond multi-class learning such as structured prediction or other tasks?
2. Could you comment on settings where assuming the data is linear separable is valid?

**Ethical Concerns:**

["NO or VERY MINOR ethics concerns only"]

**Final Justification:**

As a final score, I recommend a 5~accept. I think this work is clearly presented and provides a though provoking new result meriting it as an accepted paper.

**Limitations:**

yes

**Paper Formatting Concerns:**

No concerns.

**Quality:**

3

**Strengths And Weaknesses:**

Strengths
1. This work examines a practical setting of multiclass and finite time steps, bridging theory with practice.
2. The paper provides a clear explanation of the problem setting, allowing for a general audience to understand the technical value of this work.
3. The proofs of this work are written very clearly.

Weaknesses
1. From a theoretical standpoint, I view these results as a strong advancement; however, I would say there is a limitation from an applied standpoint, given that the setting assumes the model is a separable linear classifier.

---

> ### Author Rebuttal · Authors · 2025-07-28
>
> Thanks for the review! Below we address all of your comments; if anything remains unclear, please let us know and we will happily elaborate during the discussion period.
>
>
> > “From a theoretical standpoint, I view these results as a strong advancement; however, I would say there is a limitation from an applied standpoint, given that the setting assumes the model is a separable linear classifier.”-
>
> We appreciate the fact that you see the results as a strong theoretical advancement. The scope of this work is indeed linear models parameterized by a matrix, where even in this relatively restricted setting, the analysis and results are already non-trivial. Focus on the linear case is rather common in theoretical work in learning theory, as it allows for derivation of rigorous conclusions and concrete insights. We conjecture that some of the results in the work, e.g., the separation between $\ell_\infty$ and $\ell_2$ can be generalized to more complex regimes, and that this is a good idea for future work. We will discuss future work in more detail in the final version of the paper.
>
> > Could you comment on these insights generalizing beyond multi-class learning such as structured prediction or other tasks?
>
> This is a good point. Our current analysis focuses on multiclass prediction with a linear output layer, but we believe that some insights may extend to more general settings such as structured prediction tasks where the number of output classes is naturally very large. For example, the fact that the use of a loss function with $\ell_\infty$-smooth template like the cross entropy decreases the dependence on the number of classes, compared to other loss functions, is of particular importance in such scenarios. Extending the analysis to structured or sequence prediction is an interesting direction. We will add a brief discussion in the final version to clarify this.
>
>
> > Could you comment on settings where assuming the data is linear separable is valid?
>
> The assumption of separability is originally motivated by overparameterized models, where the number of parameters far exceeds the number of training samples. This has led to a rich line of theoretical work analyzing the performance of various algorithms under such conditions. Our work continues this line by studying linear classification under the separability assumption as a theoretical model. We hope that some of the insights derived here will be transferable to more complex and practical settings.

---

> ### Comment · Reviewer_WLTb · 2025-08-03
> **Response to rebuttal**
>
> Thank you for your reply, I find it helpful as a reviewer. As a final score, I will stay at 5~accept. I think this work is clearly presented and provides a thoughtful provoking new result meriting it as an accepted paper.

---

### Official Review · Reviewer_kY3X · 2025-07-10

**Clarity:** 3
**Significance:** 3
**Originality:** 3
**Rating:** 5
**Confidence:** 3

**Summary:**

This paper studies the implicit bias of vanilla gradient descent (GD) for multi-class classification through the generalization aspect. Similar questions have been answered for the binary classification case, indicating that generalization performance of GD is characterized by the decay rate of the loss function. This work extends the observation in binary case into multi-class case with the help of recent work on characterization of multi-class loss function via the loss template. It is shown in their main results that in the case of multi-class, the generalization performance of GD is dependent on the $L_p$ geometry of the loss template. Moreover, authors also derive a lower bound on the risk to demonstrate the tightness of their estimation.

**Questions:**

- It is mentioned that another line of work on implicit bias of GD focus on the asymptotic convergence of classifier to max-margin one. Is it possible to derive non-asymptotic estimate on that convergence?
- Some typographical errors in main text
    - in Assumption 1, $\gamma$ is strictly positive.
    - in Theorem 1, parentheses are left out for the step size
    - in Corollary 7, $\rho(x)$ should be $= x^{-\alpha}$

**Ethical Concerns:**

["NO or VERY MINOR ethics concerns only"]

**Final Justification:**

All the questions have been addressed in the authors' response, mainly about possible directions for future research. My evaluation remains positive, and I keep the rating unchanged.

**Limitations:**

yes

**Paper Formatting Concerns:**

no formatting issues found.

**Quality:**

3

**Strengths And Weaknesses:**

Strengths:
- The paper is well-written with a clear structure. Essential elements for the problem setting are properly introduced. Main statements are presented in a logical way and examples are provided to illustrate the significance of their results.
- In terms of the results, their observation provides a way to analyze loss functions with smoothness assumption in different norms in a unified way. The bounds interpolate smoothly among different exponent constant $p$.

Weakness:
The assumptions are restricted. The work would be even more compelling if it goes beyond linear case.

---

> ### Author Rebuttal · Authors · 2025-07-28
>
> Thank you for the feedback!  Below we address all of your comments; if anything remains unclear, please let us know and we will happily elaborate during the discussion period. We will of course fix all of the typos you mentioned.
>
> > “The assumptions are restricted. The work would be even more compelling if it goes beyond linear case.”
>
> The scope of this work is linear models parameterized by a matrix, where even in this relatively restricted setting, the analysis and results are already non-trivial. Focus on the linear case is rather common in theoretical work in learning theory, as it allows for derivation of rigorous conclusions and concrete insights.
>
> Investigating whether the geometry of the loss function influences the performance of gradient descent beyond the linear case is indeed an interesting direction for future work. We conjecture that the separation between different norms occurs also in more complex regimes, and therefore, some of the insights developed here may extend to broader settings. We will address these limitations and outline possible directions for future research in the next revision of the paper.
>
> > “It is mentioned that another line of work on implicit bias of GD focus on the asymptotic convergence of classifier to max-margin one. Is it possible to derive non-asymptotic estimate on that convergence?”
>
> This is a very good question. It is indeed possible to derive non-asymptotic bounds for the convergence to max-margin. However, usually the generalization bounds that one can achieve will be much worse than the optimal rates. For example, for the binary case, the output of GD satisfies $|\|w_t/||w_t|| -w^*\||=O(1/\log{T})$. This implies a very weak upper bound of $\exp(1/\gamma)$ on the number of steps till the predictor achieves a positive margin $\gamma$  on all the data points. We refer the reviewer to a more extensive discussion in page 2 of [20].

---

> > ### Comment · Reviewer_kY3X · 2025-08-05
> >
> > Thank you for your response. I do have no further questions.

---

### Official Review · Reviewer_UBJr · 2025-07-14

**Clarity:** 3
**Significance:** 2
**Originality:** 3
**Rating:** 5
**Confidence:** 3

**Summary:**

The paper considers the problem of establishing generalization loss bounds for multiclass linear classification of separable data, with finite number of gradient descent (GD) steps. Extending prior related work on binary classification to multiclass, the authors leverage the loss template framework and prove two main results - an upper bound on the population risk for a class of loss objectives, and for a special case, a lower bound on the risk, making the bound tight.

Overall, I would be inclined positively, if the authors clarify some of my questions.

**Questions:**

Q1. In Cor. 7, do the authors intend to say $\rho(x) = x^{-\alpha}$, instead of $\exp^{-x}$? The discussion is on polynomial decay, likely a typo.
Q2. From line 312, could the authors elaborate on “This differ but not at odds with the results of [15], which suggest that exponentially tailed losses exhibit similar asymptotic behavior”, for my understanding? Specifically, what does the result of Ravi et al, referenced here say and how is the current result not at odds with the same? Does the setup differ?
Q3. While separability is an assumption, it would be helpful to see any insights on what conditions are needed for the data to be linearly separable in a multi-class sense. In particular, with an interest by authors on increasing $k$, I imagine that separability would be harder with an increasing number of classes, under a certain data model. For example, Kini and Thrampoulidis, 2021, studied conditions for binarized multiclass separability. Is there an understanding for the separability criterion considered by the authors? Does data dimension play a role in the risk bounds?


Kini and Thrampoulidis, Phase Transitions for One-Vs-One and One-Vs-All Linear Separability in Multiclass Gaussian Mixtures, 2021

**Ethical Concerns:**

["NO or VERY MINOR ethics concerns only"]

**Final Justification:**

I believe the paper provides new theoretical results to improve our understanding of multi class classification under a separability assumption. Reason for not giving a higher rating is that the practical applicability may be limited due to the strong assumption, which future works may try and alleviate.

**Limitations:**

The work is theoretical and addresses a specific question under strong data and modeling assumptions. The authors do not discuss societal impact as such for the work, which is understandable. The limitations from the perspective of assumptions is mentioned in the checklist. It might be a good idea to note this in the main body of the paper.

**Quality:**

3

**Strengths And Weaknesses:**

Strengths

S1. From my understanding, the results are novel for the considered setup of multiclass linear classification of separable data with finite number of GD steps. The setup is also wide enough to cover loss functions that decay to 0 on separable data, with example applications shown by the authors on cross-entropy and polynomial loss.
S2. While I have not checked the proofs for underlying lemmas in detail, the proof techniques are based on established sound frameworks and are novel extensions.
S3. In particular, the bounds show a clear difference between two sub-classes of loss objectives, that of functions smooth with respect to $\infinity$ and 2-norms, showing dependence on the number of classes. is notable. I would be interested to see if the results indicate any implications on choice of loss functions for extreme classification (large number of classes)? In the generative modeling setup, for example, do we gain some understanding when the vocabulary sizes are very large?

Weaknesses:

W1. I would like to see a broader discussion of the applications of the results, given the assumptions of separable data, linear classification etc, especially given strong data and modeling assumptions, whose applicability may be questioned by some audience. For example, do the results help understand properties of classifier-head fine-tuning in deep learning? A meaningful discussion along this direction would help contextualize the paper for a broader audience.
W2. The writing is generally well done, especially on notations and setup. However, I found that the flow can be improved - for instance, the paper ends rather abruptly at an example application of one of the results to a special case of polynomial tailed loss. Related to the point above, the results could benefit from a practical discussion. Further, what do the authors consider as problems that future work may address?

---

> ### Author Rebuttal · Authors · 2025-07-28
>
> Thanks for the review! Below we address all of your comments; if anything remains unclear, please let us know and we will happily elaborate during the discussion period.
>
> > “I would be interested to see if the results indicate any implications on choice of loss functions for extreme classification (large number of classes)? In the generative modeling setup, for example, do we gain some understanding when the vocabulary sizes are very large?”
>
> This is an excellent point. While our classification result is formally established in the linear setting, we conjecture that some of its insights may extend to more general scenarios. For example, it may help explain the common use of cross-entropy loss in more complex models. According to our bounds, a loss function with $\ell_\infty$-smooth template like the cross-entropy reduces the dependence on the number of classes, which, in extreme classification tasks with a very large number of classes, can result in significant differences in performance compared to other loss functions.
>
> >” I would like to see a broader discussion of the applications of the results, given the assumptions of separable data, linear classification etc,... For example, do the results help understand properties of classifier-head fine-tuning in deep learning? A meaningful discussion along this direction would help contextualize the paper for a broader audience.”
>
> Following up on our previous response, classifier-head fine-tuning (assuming you meant the last layer of a pretrained model) is yet another example of a multi-class prediction problem with a large number of classes, where our results give support for using a loss function $\ell_\infty$-smooth template such as the cross-entropy loss. We agree that a discussion about these implications and applications is warranted, and we will include one in the final version.
>
>
>
>
> >  “the paper ends rather abruptly at an example application of one of the results... the results could benefit from a practical discussion. Further, what do the authors consider as problems that future work may address?”
>
> We agree - we will discuss the implications and limitations of our work, as well as directions for future work, in the final version of the paper. Future research may study whether the geometry of the loss function influences the behavior of gradient descent beyond the linear setting. We conjecture that the separation between different norms occurs in more complex regimes.
>
> > “Typo in corollary 7:”
>
> Thanks for pointing that out. We will fix this in the next revision.
>
> >  “What does the result of Ravi et al, referenced here say and how is the current result not at odds with the same? Does the setup differ?”
>
> Reference [15] analyzes a setup similar yet different from ours. They show that for any loss function that decays exponentially to zero, gradient descent exhibits similar asymptotic behavior and converges in direction to the max-margin solution. In contrast, our work deals with more general loss functions and reaches a different conclusion, demonstrating that finite-time generalization depends on the geometry of the loss function. Despite this difference, our results are not in conflict with [15], as we deal with a different setup and we further focus on different aspects: finite-time behavior versus asymptotic behavior.
>
> >” it would be helpful to see any insights on what conditions are needed for the data to be linearly separable in a multi-class sense… Is there an understanding for the separability criterion considered by the authors? Does data dimension play a role in the risk bounds?”
>
> The separability criterion considered in this work is *One-Vs-All linear separability*. This assumption is originally motivated by overparameterized neural networks, where the number of parameters significantly exceeds the number of examples $n$. In our setting, the model has $kd$ parameters, where $d$ is the dimension of the data. Since our bounds depend on $k$ but not on $d$, this suggests that when $d$ is large, the data may become separable without negatively impacting the performance of gradient descent.
>
> >”The limitations from the perspective of assumptions is mentioned in the checklist. It might be a good idea to note this in the main body of the paper.”
>
> Thanks for this advice. We will add a discussion about the limitations of our work in the next revision of the paper.

---

> > ### Comment · Reviewer_UBJr · 2025-08-08
> > **Thanks for the response**
> >
> > I thank the author's for the response. Adding the discussion above would help the paper in my opinion. I will update my rating to 5.

---

### Decision · Program_Chairs · 2025-09-17

**Decision:**

Accept (poster)

**Comment:**

This is a strong theoretical contribution where reviewers found the contirbutions novel, rigorous, and clearly presented. While the assumption of linear separability limits direct applicability, the results advance our theoretical understanding of multiclass classification and offer useful insights for loss design in extreme classification.